# Self-Supervised Laplace Approximation for Bayesian Uncertainty Quantification

**Julian Rodemann**  *julian.rodemann@cispa.de*
*Rational Intelligence Lab, CISPA Helmholtz Center for Information Security*
*Department of Statistics, LMU Munich*

**Alexander Marquard**  *alexander.marquard@campus.lmu.de*
**Thomas Augustin**  *thomas@stat.uni-muenchen.de*
*Department of Statistics, LMU Munich*

**Michele Caprio**  *michele.caprio@manchester.ac.uk*
*Department of Computer Science, The University of Manchester*

**Reviewed on OpenReview:** *https://openreview.net/forum?id=T8w8L2t3JG*

## Abstract

Approximate Bayesian inference typically revolves around computing the posterior parameter distribution. In practice, however, the main object of interest is often a model's *predictions* rather than its parameters. In this work, we propose to bypass the parameter posterior and focus directly on approximating the posterior predictive distribution. We achieve this by drawing inspiration from self-training within self-supervised and semi-supervised learning. Essentially, we quantify a Bayesian model's predictive uncertainty by refitting on self-predicted data. The idea is strikingly simple: If a model assigns high likelihood to self-predicted data, these predictions are of low uncertainty, and vice versa. This yields a deterministic, sampling-free approximation of the posterior predictive. The modular structure of our Self-Supervised Laplace Approximation (SSLA) further allows us to plug in different prior specifications, enabling classical Bayesian sensitivity (w.r.t. prior choice) analysis. In order to bypass expensive refitting, we further introduce an approximate version of SSLA, called ASSLA. We study (A)SSLA both theoretically and empirically in regression models ranging from Bayesian linear models to Bayesian neural networks. Across a wide array of regression tasks with simulated and real-world datasets, our methods outperform classical Laplace approximations in predictive calibration while remaining computationally efficient.

## 1 Introduction

Despite all the merits of Bayesian methods, one of their notorious shortcomings is the fact that in many applications the posterior turns out to be computationally intractable. Let $\Theta$ denote the parameter space of interest and $\mathcal{S}$ denote the sample space. Let $\theta$ denote a generic element of $\Theta$, and $D$ be a collection of elements of $\mathcal{S}$ corresponding to the collected evidence. By Bayes' theorem, we have that

$$p(\theta \mid D) = \frac{p(D \mid \theta)\pi(\theta)}{p(D)} = \frac{p(D \mid \theta)\pi(\theta)}{\int_{\Theta} p(D \mid \theta)\pi(\theta)\mathrm{d}\theta}, \tag{1}$$

where for maximum generality we assumed $\Theta$ and $\mathcal{S}$ to be uncountable, so that $p(\theta \mid D)$ is the probability density function (pdf) of posterior $P(\cdot \mid D) \equiv P_D$, $p(D \mid \theta)$ is the pdf of likelihood $P_\theta$, and $\pi(\theta)$ is the pdf of prior $\Pi$.[1] Oftentimes, the denominator in equation 1 can only be solved numerically. As a consequence, to

---

[1]Notice that $p(D)$ is the pdf of the *marginal likelihood*.

obtain the posterior, the scholar needs to resort to approximation techniques such as Variational Inference (VI), Markov Chain Monte Carlo (MCMC) or Laplace approximations, see Section 2.

Usually, these approximations require a large computational overhead, which makes Bayesian techniques slower than their frequentist counterparts. This is especially true in Machine Learning (ML) and Artificial Intelligence (AI) applications, where Bayesian Neural Networks (BNNs) are much slower to train than classical NNs. In this work, we propose a technique to approximate the posterior predictive $p(\hat{D} \mid \theta, D)$, where $\hat{D}$ denotes unseen test data, that allows us to overcome this type of shortcoming. Loosely inspired by recent work on martingale posteriors (Fong et al., 2023; Lee et al., 2023; Moya & Walker, 2025), we forego explicit calculation of the (parameter) posterior and focus on the posterior predictive distribution. The posterior predictive is a crucial quantity in practical Bayesian inference: It informs the scholar about the distribution of the predictions resulting from the posterior distribution of $\theta$. It constitutes one major, if not *the main* advantage of Bayesian inference over frequentist procedures. The latter lack inherent methods of quantifying predictive uncertainty, because $\theta$ is not treated as a random variable.

Our approximation is based on the well-known Laplace approximation method; this means that the posterior predictive is approximated by an unnormalized Gaussian distribution. We note in passing that this is not necessarily a severely restrictive assumption: For example, it is customary in Variational Inference (VI) to approximate the true posterior with its closest (according to the KL divergence) Gaussian distribution.

The proposed self-supervised Laplace approximation is based on a simple, yet far-reaching insight from reciprocal learning (Rodemann et al., 2024; Rodemann & Bailie, 2025), or more specifically, from self- and semi-supervised learning (Chapelle et al., 2006; Zhu & Goldberg, 2009; Jing & Tian, 2021): We can learn something from refitting on self-predicted data. Contrary to self- and semi-supervised learning, however, we do not aim at increasing predictive accuracy, but approximate the predictions' uncertainty.

Augmenting a likelihood with an additional (pseudo-)observation is a standard Bayesian device under conditional independence assumptions. Our contribution is *not* the use of pseudo-observations per se, but the resulting *closed-form Laplace approximation of the posterior predictive* evaluated at the model's self-prediction. Concretely, we derive a tractable approximation of $\log p(\hat{y}_{n+1} \mid x_{n+1}, D)$ that decomposes into a difference of three interpretable increments: (i) a log-likelihood increment, (ii) a log-prior increment, and (iii) a curvature (log-determinant) increment. This decomposition yields a practical predictive *sensitivity signal*: the larger the change induced by $\hat{y}_{n+1}$, the higher the predictive uncertainty assigned to that prediction. To avoid ambiguity with broad uses of the expression "self-supervised learning", we interpret the construction as a self-training based Laplace approximation and refer to the method accordingly.

In a nutshell, we propose to quantify uncertainty of predictions by refitting on these very predictions. Concretely, we investigate how a model's likelihood/loss changes if we *ceteris paribus* refit the model on the enhanced data set. Intuitively, the lower the model's likelihood of self-predicted data, the higher the model's predictive uncertainty.

The remainder of this article is structured as follows. In Section 2, we briefly review key methods in approximate Bayesian inference. Section 3 introduces our Self-Supervised Laplace Approximation (SSLA) method, detailing the theoretical foundations. All proofs are relegated to Appendix A. Section 4 presents extensive experiments: First, we validate our approximation on classical conjugate-prior models for regression problems, then we demonstrate the method's performance in heteroscedastic regression tasks using neural networks, comparing against established Bayesian inference approaches. We conclude with evaluations on various real-world datasets, underlining practical applicability and robustness of our approach. Section 5 concludes the paper, summarizing findings and potential directions for further research.

## 2 Background and Related Work

Approximate Bayesian Inference (ABI) tackles the practical issue of computing posterior distributions that are typically analytically intractable. In particular, the central tools in Bayesian Deep Learning (BDL), viz. Bayesian Neural Networks (BNNs), often face significant computational challenges, necessitating efficient approximations. This section briefly discusses BNNs, and reviews key approximation techniques, including Variational Inference, Markov Chain Monte Carlo (MCMC), and Laplace.

**Bayesian Neural Networks.** BNNs incorporate Bayesian inference into neural network training by modeling the network weights probabilistically. Prominent approaches involve explicitly setting priors on weights (Blundell et al., 2015) or implicitly via dropout techniques (Gal & Ghahramani, 2016), interpreting dropout as a form of approximate Bayesian inference. Techniques like Stochastic Gradient Langevin Dynamics (Welling & Teh, 2011) provide scalable inference options.

**Variational Inference.** Variational Inference (VI) approximates the posterior distribution through optimization, minimizing the KL divergence between the true posterior and a set of "simpler" distributions (Blei et al., 2017). Classical methods include mean-field variational Bayes (MFVB) (Beal, 2003) and importance-weighted autoencoders (IWAE) (Mnih & Gregor, 2014). Deep generative models, notably Variational Autoencoders (VAEs) (Kingma & Welling, 2014), extended these frameworks, incorporating hierarchical modeling (Higgins et al., 2017) and amortized inference (Kingma & Welling, 2014). Techniques such as stochastic variational inference (SVI) (Beal & Ghahramani, 2000) and black-box variational inference (BBVI) (Ranganath et al., 2014) improve scalability but struggle to keep pace with increasingly complex architectures. Ortega et al. (2024) merge variational inference and Laplace approximations by a variational sparse Gaussian Process approach, enabling sub-linear training time.

**Markov Chain Monte Carlo (MCMC).** MCMC techniques provide theoretically exact posterior sampling by constructing Markov chains whose stationary distributions represent the (true) posterior. Methods such as Hamiltonian Monte Carlo (Neal, 1993) and the No-U-Turn Sampler (NUTS) (Hoffman & Gelman, 2014) are widely employed. However, computational demands limit their practicality in large-scale scenarios (Papamarkou et al., 2022), partly addressed by Wiese et al. (2023); Sommer et al. (2024; 2025).

**Laplace Approximation.** Laplace approximations efficiently approximate posterior distributions with Gaussian distributions centered at the posterior mode, leveraging the posterior's local curvature. Recent advances, such as Kronecker-factored Hessian approximations (Eschenhagen et al., 2023), improve computational efficiency significantly, see also Antoran et al. (2022); Bouchiat et al. (2023); Daxberger et al. (2021a;b); Eschenhagen et al. (2023); Immer et al. (2021b); Cinquin et al. (2024). Integrated Nested Laplace Approximations (INLA) (Rue et al., 2009; Martino & Riebler, 2019) extend these methods to handle multimodal posterior landscapes more robustly.

The basis for Laplace approximation is Laplace's *method*, introduced by Pierre-Simon de Laplace in 1774 (Laplace, 1986). It serves as a tool for approximating integrals of the form

$$\int_a^b e^{Cf(x)} \, \mathrm{d}x, \tag{2}$$

where $f(x)$ is a twice-differentiable function, $C > 0$ is a constant, and the boundaries $a$ and $b$ may potentially extend to infinity. In the context of Bayesian statistics, Laplace approximation usually denotes either the estimation of the normalizing constant $\int_\Theta p(D \mid \theta)\pi(\theta)\mathrm{d}\theta$ (marginal likelihood) (Llorente et al., 2023) using Laplace's method, or the approximation of the posterior distribution $p(\theta \mid D)$ (Tierney & Kadane, 1986) through a Gaussian centered at the maximum a posteriori estimate, see Laplace (1986); Schwarz (1978); Tierney & Kadane (1986) for initial works and Konishi & Kitagawa (2008); Llorente et al. (2023); Turkman et al. (2019) for modern textbook proofs of the approximation's properties. The Laplace approximation $\breve{p}_L(\theta \mid D)$ of $p(\theta \mid D)$ builds on a second-order Taylor approximation of the likelihood, which then yields a Gaussian integral (Gauß, 1877), whose solution gives

$$p(\theta \mid D) \approx \breve{p}_L(\theta \mid D) := (2\pi)^{-q/2} \left| H_{\mathrm{post}}(\hat{\theta}_{\mathrm{MAP}}) \right|^{1/2} \exp\left( -\tfrac{1}{2}(\theta - \hat{\theta}_{\mathrm{MAP}})^\top H_{\mathrm{post}}(\hat{\theta}_{\mathrm{MAP}})(\theta - \hat{\theta}_{\mathrm{MAP}}) \right), \tag{3}$$

where $q$ is the dimension of $\Theta$, $\hat{\theta}_{\mathrm{MAP}}$ the MAP estimator $\hat{\theta}_{\mathrm{MAP}} := \arg\max_\theta \left\{ \ell_D(\theta) + \log\pi(\theta) \right\}$, and $H_{\mathrm{post}}(\hat{\theta}_{\mathrm{MAP}}) := -\nabla_\theta^2 \left( \ell_D(\theta) + \log\pi(\theta) \right)\big|_{\theta=\hat{\theta}_{\mathrm{MAP}}}$ the negative Hessian of the log-posterior at $\hat{\theta}_{\mathrm{MAP}}$. In density form, $p(\theta \mid D) \approx \mathcal{N}\big(\theta; \hat{\theta}_{\mathrm{MAP}}, H_{\mathrm{post}}(\hat{\theta}_{\mathrm{MAP}})^{-1}\big)$. The computation of the Hessian was long considered (and to some degree, still is) a computational bottleneck in deploying Laplace approximations. A direct computation grows quadratically in $d$. The Gauss-Newton approximation has long been the most popular way to circumvent such computational hurdle (Ritter et al., 2018b;a; Kristiadi et al., 2020; Lee et al., 2020; Immer et al.,

2021b). Recently, however, the Kronecker-factored approximate curvature (KFAC) has attracted increasing attention due to its scalability and simplicity (Eschenhagen et al., 2023). KFAC relies on block-diagonal factorization of the Hessian (Martens & Grosse, 2015).

In this work, we shift the focus to the posterior predictive distribution

$$p(\hat{y}_{n+1} \mid x_{n+1}, D) = \int_{\Theta} p(\hat{y}_{n+1} \mid x_{n+1}, \theta) \, p(\theta \mid D) \, \mathrm{d}\theta, \tag{4}$$

where $x_{n+1}$ is a new input (feature) and $\hat{y}_{n+1}$ is the predicted output (target) with $n$ the cardinality of the collected evidence $D$. While harder to approximate (at least at first sight, see Section 3), the posterior predictive is the most relevant quantity for practical prediction problems addressed via a Bayesian methodology.

It provides the user with direct information on the uncertainty of the predictions, rather than on the parameter space. The *modus operandi* in deploying Laplace approximations in BNNs is to approximate the posterior $p(\theta \mid D)$ by some Laplace approximation $\check{p}_L(\theta \mid D)$ and then use plug-in MC-samples from

$$\int_{\Theta} p(\hat{y}_{n+1} \mid x_{n+1}, \theta) \, \check{p}_L(\theta \mid D) \, \mathrm{d}\theta \tag{5}$$

to approximate the posterior predictive distribution (Bosch et al., 2022; Kristiadi et al., 2022). The necessity of MC-sampling drastically increases the computational overhead. Daxberger et al. (2021a) summarize and implement a few alternatives ranging from probit approximation (Spiegelhalter & Lauritzen, 1990) for binary classification to extended probit (Gibbs, 1998) or approximating the softmax-Gaussian integral via a Dirichlet distribution (Hobbhahn et al., 2022) for general classification.

**This is the regime we address in this work:** we derive a direct Laplace approximation of the posterior predictive that avoids Monte Carlo integration while remaining applicable to generic deep regression models. We position (A)SSLA as a complementary approach to reduce predictive sampling cost, while acknowledging that specialized alternatives exist in particular regimes (e.g., moment-propagation methods, deterministic/low-variance VI variants, and Bayesian last-layer models).

In addition, our method is modular with respect to the prior specification. That is, it can incorporate different priors directly in the approximation. This is especially important in the newly-introduced field of Credal Bayesian Deep Learning (CBDL, Caprio et al. (2024)) building on the generalized Bayes rule by Walley (1991), see also Walter & Augustin (2009); Rodemann et al. (2023a). There, finite collections of priors and of likelihoods are combined pairwise (thus inducing a combinatorially expensive problem to solve) and VI-approximated to derive a finite set of posteriors, and in turn a finite set of posterior predictives. The convex hull of the latter forms a credal set, i.e. a closed and convex set of probabilities, a central concept in Imprecise Probability theory (Walley, 1991; Augustin et al., 2014; Troffaes & de Cooman, 2014).[2]

While there is currently no broadly adopted deterministic method that eliminates the need for MC sampling in modern Bayesian neural networks, there are alternatives for special settings that can reduce or avoid brute-force Monte Carlo (MC) sampling at prediction time in certain settings. These methods are not (yet) a broadly adopted, general-purpose replacement for MC integration for posterior predictives in modern deep Bayesian regression models. Examples comprise approaches such as probabilistic backpropagation that propagate approximate moments through the network to obtain predictive uncertainties without explicit MC sampling. They rely on strong distributional and independence approximations and historically have not become the default choice for large-scale modern architectures (Hernández-Lobato & Adams, 2015). Moreover, there exist deterministic (or low-variance) alternatives to standard stochastic variational inference that aim to approximate predictive moments with reduced sampling noise; these are nonetheless specialized approximations and are not the community-default for obtaining posterior predictives in deep regression BNNs (Wu et al., 2018). Furthermore, restricting uncertainty to the final layer can yield analytic or near-analytic regression predictives, but this constitutes a different modeling choice rather than a general solution

---

[2]For more modern references, see e.g. Caprio & Mukherjee (2023); Caprio & Seidenfeld (2023); Lu et al. (2024); Caprio et al. (2025); Dutta et al. (2025); Caprio (2025); Rodemann & Augustin (2021; 2022; 2024); Jansen et al. (2022a;b; 2023; 2024a); Jansen (2025); Bailie & Derr (2025); Fröhlich (2025).

for full-network Bayesian regression posterior predictive evaluation (Watson et al., 2021; Harrison et al., 2024).

# 3 Self-Supervised Laplace Approximation Inspired By Self-Training

In this section, we introduce Self-Supervised Laplace Approximation (SSLA), as well as the idea for its approximate counterpart. As pointed out before, SSLA bypasses the explicit computation of the (parameter) posterior distribution, focusing directly on approximating the posterior predictive distribution. This is achieved by refitting on self-predicted data, drawing inspiration from self-supervised and semi-supervised learning. Specifically, we draw inspiration from approximately Bayes-optimal pseudo-labeling in semi-supervised learning, where the likelihood is jointly evaluated for labeled and (new) self-labeled data (Rodemann et al., 2023b;c), see also Rodemann (2023; 2024); Dietrich et al. (2024) and from related work on cautious disambiguation of set-valued labels (Rodemann et al., 2022). As it turns out, SSLA allows us to plug-in and plug-out priors in a modular fashion. Further approximations allow us to get rid of the requirement to refit the model on self-predicted data. We call this version the Approximate Self-Supervised Laplace Approximation (ASSLA) and introduce it formally in Section 4.

Let $\mathcal{S} = \mathcal{X} \times \mathcal{Y}$, where $\mathcal{X}$ is the space of inputs and $\mathcal{Y}$ is the space of outputs. Suppose we observed $n$ data points $D = \{(x_i, y_i)\}_{i=1}^n \subset \mathcal{S}$, and assume that the (conditional) likelihood is given by $\prod_{i=1}^n p(y_i \mid x_i, \theta)$.[3] Further, let $f_\theta(x_{n+1})$ be some (frequentist) base model that is parameterized by $\theta$ and produces predictions $\hat{y}_{n+1}$ given new observations $x_{n+1}$. We call $D_{n+1} = \{(x_i, y_i)\}_{i=1}^n \cup \{(x_{n+1}, \hat{y}_{n+1})\}$ the augmented dataset. Consider the (posterior) predictive distribution, i.e., the distribution of any Bayesian model's predictions $(x_{n+1}, \hat{y}_{n+1})$ given training data $D$,

$$p(\hat{y}_{n+1} \mid x_{n+1}, D) = \int_\Theta p(\hat{y}_{n+1} \mid x_{n+1}, \theta)\, p(\theta \mid D)\, \mathrm{d}\theta. \tag{6}$$

The main idea behind our approximation is to expand the likelihood $p(D \mid \theta)$ in equation 1 using the likelihood of the model's prediction $p(\hat{y}_{n+1} \mid x_{n+1}, \theta)$. This allows us to forgo the need to compute $p(\theta \mid D)$ entirely. Define $\ell_D(\theta) := \log p(D \mid \theta) = \sum_{i=1}^n \log p(y_i \mid x_i, \theta)$ as the classical log-likelihood of $D$, and $\ell_{(x_{n+1}, \hat{y}_{n+1})}(\theta) := \log p(\hat{y}_{n+1} \mid x_{n+1}, \theta)$ as the log-likelihood of the predicted instance. Further, denote their sum as $\tilde{\ell}(\theta) := \ell_{(x_{n+1}, \hat{y}_{n+1})}(\theta) + \ell_D(\theta)$. As a consequence of Bayes' theorem (equation 1), we can write the integrand in equation 6 as

$$p(\hat{y}_{n+1} \mid x_{n+1}, \theta)\, p(\theta \mid D) = \frac{\exp\left[\tilde{\ell}(\theta)\right] \pi(\theta)}{p(D)}.$$

Let $\mathcal{J}(\theta) = -\boldsymbol{H}(\ell_D(\theta)) = -\nabla_\theta^2 \ell_D(\theta)$ denote the observed Fisher information matrix (the negative Hessian) and $\tilde{\mathcal{J}}(\theta) = -\nabla_\theta^2 \tilde{\ell}(\theta)$ the observed Fisher information matrix of the augmented dataset, respectively. Further denote by $\tilde{\theta} \in \arg\max_\theta \tilde{\ell}(\theta)$ the maximizer of $\tilde{\ell}(\theta)$. It holds that $\left.\frac{\partial \tilde{\ell}(\theta)}{\partial \theta}\right|_{\theta = \tilde{\theta}} = 0$ by definition of $\tilde{\theta}$. A Taylor expansion of the second order around $\tilde{\theta}$ thus gives

$$\tilde{\ell}(\theta) \approx \tilde{\ell}(\tilde{\theta}) - \frac{1}{2}(\theta - \tilde{\theta})^\top \tilde{\mathcal{J}}(\tilde{\theta})(\theta - \tilde{\theta}).$$

The integrand decays exponentially in $\|\theta - \tilde{\theta}\|_2$, where $\|\cdot\|_2$ denotes the Euclidean norm.[4] This allows us to approximate it locally around $\tilde{\theta}$ by $\pi(\theta) \approx \pi(\tilde{\theta})$ inside the integral with another Taylor series. We refer to (Miller, 2006, Section 3.7) for a detailed treatment of the remainder terms and regularity conditions; Specifically, see (Łapiński, 2019, Theorem 2) for background on $\pi(\theta) \approx \pi(\tilde{\theta})$. We thus approximate $p(\hat{y}_{n+1} \mid x_{n+1}, D)$ by

$$\frac{\exp\left[\tilde{\ell}(\tilde{\theta})\right] \pi(\tilde{\theta})}{p(D)} \int_\Theta \exp\left[-\frac{1}{2}(\theta - \tilde{\theta})^\top \tilde{\mathcal{J}}(\tilde{\theta})(\theta - \tilde{\theta})\right] \mathrm{d}\theta. \tag{7}$$

---

[3]As is customary, we assume that given inputs $x_i$ and parameter $\theta$, the $Y_i$ have densities $p(y_i \mid x_i, \theta)$ and are independent across $i$.

[4]This implies that $\Theta$ is a subset of a Euclidean space $\mathbb{R}^q$. If instead $\Theta$ is a subset of a generic normed vector space $(V, \|\cdot\|_V)$, substitute $\|\cdot\|_2$ with $\|\cdot\|_V$.

The integral $\int_{\Theta} \exp\left[-\frac{1}{2}(\theta - \tilde{\theta})^{\top} \tilde{\mathcal{J}}(\tilde{\theta})(\theta - \tilde{\theta})\right] \mathrm{d}\theta$ is a Gaussian integral (Gauß, 1877). Defining $\Sigma := [\tilde{\mathcal{J}}(\tilde{\theta})]^{-1}$ and $\phi_{\Sigma}$ as the density of the multivariate Normal distribution $\mathcal{N}(0, \Sigma)$, we have that

$$\int_{\Theta} \exp\left[-\frac{1}{2}(\theta - \tilde{\theta})^{\top} \tilde{\mathcal{J}}(\tilde{\theta})(\theta - \tilde{\theta})\right] \mathrm{d}\theta = (2\pi)^{q/2} |\Sigma|^{1/2} \int_{\Theta} \phi_{\Sigma}(\theta) \mathrm{d}\theta = (2\pi)^{q/2} |\tilde{\mathcal{J}}(\tilde{\theta})|^{-1/2}, \tag{8}$$

where $q$ is the dimension of the Euclidean space we are working in, and $|\cdot|$ denotes the determinant operator.[5] Combining equation 7 and equation 8, we obtain

$$p(\hat{y}_{n+1} \mid x_{n+1}, D) \approx (2\pi)^{q/2} \frac{\exp\left[\tilde{\ell}(\tilde{\theta})\right]\pi(\tilde{\theta})}{|\tilde{\mathcal{J}}(\tilde{\theta})|^{1/2}p(D)}. \tag{9}$$

Taking the logarithm of equation 9, we get

$$\log p(\hat{y}_{n+1} \mid x_{n+1}, D) \approx \frac{q}{2} \log(2\pi) + \tilde{\ell}(\tilde{\theta}) + \log \pi(\tilde{\theta}) - \frac{1}{2} \log |\tilde{\mathcal{J}}(\tilde{\theta})| - \log p(D). \tag{10}$$

Borrowing from some classical Laplace approximations of the marginal likelihood (Bishop & Nasrabadi, 2006; Konishi & Kitagawa, 2008; Llorente et al., 2023; Schwarz, 1978), we can approximate $\log p(D)$ as follows

$$\begin{aligned}
\log p(D) &= \log \int_{\Theta} p(D \mid \theta) \, \pi(\theta) \, \mathrm{d}\theta \\
&\approx \cdots = \ell_D(\hat{\theta}) + \frac{q}{2} \log(2\pi) - \frac{1}{2} \log |\mathcal{J}(\hat{\theta})| + \log \pi(\hat{\theta}),
\end{aligned} \tag{11}$$

where $\hat{\theta} \in \arg\max_{\theta} \ell_D(\theta)$, see Konishi & Kitagawa (2008, Section 9.1.3). Combining equation 10 and equation 11, we obtain

$$\log p(\hat{y}_{n+1} \mid x_{n+1}, D) \approx \tilde{\ell}(\tilde{\theta}) - \ell(\hat{\theta}) + \log \pi(\tilde{\theta}) - \log \pi(\hat{\theta}) - \frac{1}{2} \log |\tilde{\mathcal{J}}(\tilde{\theta})| + \frac{1}{2} \log |\mathcal{J}(\hat{\theta})|. \tag{12}$$

**Equation 12 embodies the main idea behind Self-Supervised Laplace:** We approximate the posterior predictive at $\hat{y}_{n+1}$ by the change this very $\hat{y}_{n+1}$ causes in the model fit's prior, likelihood, and Fisher information (i.e., all evaluated at the model fit). Note that these changes are driven by a change in parameter (from $\hat{\theta}$ to $\tilde{\theta}$) as well as in functional form (from $\ell$ to $\tilde{\ell}$ and from $\mathcal{J}$ to $\tilde{\mathcal{J}}$, respectively). Equation 12 also shows the modularity of our approximation. As the effect of the prior is additive, no refitting under different prior specifications is required. In Appendix E, we expand on and illustrate such a prior modularity.

**On (non-)independence of the pseudo-observation:** We emphasize that we defined $\tilde{\ell}(\tilde{\theta}) = \ell(\tilde{\theta}) + \ell_{(x_{n+1}, y_{n+1})}(\tilde{\theta})$, which implies that the log-likelihood must be evaluated twice—once for the existing data $D$, and once for the new observation $(x_{n+1}, \hat{y}_{n+1})$. If we instead wanted to evaluate the likelihood only once—namely, only for the combined dataset $D \cup \{(x_{n+1}, \hat{y}_{n+1})\}$—we would have to assume independence between $D$ and $\{(x_{n+1}, \hat{y}_{n+1})\}$. However, since $\hat{y}_{n+1}$ is derived as a function of the dataset $D$, this assumption of independence does not hold. The same reasoning applies to the extended Fisher info $\tilde{\mathcal{J}}(\tilde{\theta})$. We thus abstain from such a simplification and provide further approximation instead. To sum up, our approximation does *not* require treating $(x_{n+1}, \hat{y}_{n+1})$ as an independent data point. Indeed, since $\hat{y}_{n+1}$ is itself a function of $D$, any assumption of conditional independence would be incorrect.

**Implementation summary:** Algorithm 1 summarizes SSLA, which uses the augmented objective and evaluates predictive uncertainty via likelihood, prior, and curvature increments. Algorithm 2 summarizes ASSLA, which replaces $\tilde{\theta}$ with $\hat{\theta}$ and retaining the leading-order likelihood and curvature effects, see Section 4 right below. In both cases, the augmented objective is used as a sensitivity probe and is not interpreted as the likelihood of an enlarged i.i.d. dataset.

---

[5]In the general case, $q = \dim(V)$.

---

**Algorithm 1** SSLA: Self-Supervised Laplace Approximation of the Posterior Predictive

---

**Input:** Dataset $D = \{(x_i, y_i)\}_{i=1}^n$, test input $x_{n+1}$, prior $\pi(\theta)$, log-likelihood $\log p(y \mid x, \theta)$, training objective
$\ell_D(\theta) = \sum_{i=1}^n \log p(y_i \mid x_i, \theta)$.

**Output:** Approximate log-posterior-predictive $\log \hat{p}_{\text{SSLA}}(y \mid x_{n+1}, D)$.

1: **Fit on training data:** $\hat{\theta} \in \arg\max_\theta \{\ell_D(\theta) + \log \pi(\theta)\}$.
2: **Self-prediction:** choose $\hat{y}_{n+1}$ as the point prediction at $x_{n+1}$ under $\hat{\theta}$ (e.g. predictive mean / MAP prediction).
3: **Define** $\tilde{\ell}(\theta) \leftarrow \ell_D(\theta) + \log p(\hat{y}_{n+1} \mid x_{n+1}, \theta)$.
4: **Obtain** $\tilde{\theta} \in \arg\max_\theta \{\tilde{\ell}(\theta) + \log \pi(\theta)\}$.
5: **Curvatures:**
6:     $J(\hat{\theta}) \leftarrow -\nabla_\theta^2 (\ell_D(\theta) + \log \pi(\theta))\big|_{\theta=\hat{\theta}}$
7:     $\tilde{J}(\tilde{\theta}) \leftarrow -\nabla_\theta^2 (\tilde{\ell}(\theta) + \log \pi(\theta))\big|_{\theta=\tilde{\theta}}$
8: **Compute SSLA log-PPD (up to constant):**
9:     $\log \hat{p}_{\text{SSLA}}(y \mid x_{n+1}, D) \leftarrow [\ell_D(\tilde{\theta}) - \ell_D(\hat{\theta})] + [\log \pi(\tilde{\theta}) - \log \pi(\hat{\theta})]$
10:          $+ \log p(y \mid x_{n+1}, \tilde{\theta}) - \frac{1}{2}[\log |\tilde{J}(\tilde{\theta})| - \log |J(\hat{\theta})|]$.
11: **return** $\log \hat{p}_{\text{SSLA}}(y \mid x_{n+1}, D)$.

---

**Algorithm 2** ASSLA: Approximation of SSLA

---

**Input:** Dataset $D = \{(x_i, y_i)\}_{i=1}^n$, test input $x_{n+1}$, prior $\pi(\theta)$, log-likelihood $\log p(y \mid x, \theta)$, fitted parameter $\hat{\theta}$, curvature $J(\hat{\theta})$.

**Output:** Approximate log-posterior-predictive $\log \hat{p}_{\text{ASSLA}}(y \mid x_{n+1}, D)$ (up to an additive constant independent of $y$).

1: **Self-prediction:** choose $\hat{y}_{n+1}$ as the point prediction at $x_{n+1}$ under $\hat{\theta}$.
2: **Set** $\tilde{\theta} \leftarrow \hat{\theta}$ (ASSLA's defining approximation).
3: **Local increment terms at $\hat{\theta}$:**
4:     $\Delta_\ell(y) \leftarrow \log p(y \mid x_{n+1}, \hat{\theta}) - \log p(\hat{y}_{n+1} \mid x_{n+1}, \hat{\theta})$
5:     $\Delta_J(y) \leftarrow \log |J(\hat{\theta}) + J_{n+1}(y; \hat{\theta})| - \log |J(\hat{\theta})|$
6:      where $J_{n+1}(y; \hat{\theta})$ denotes the observed curvature contribution from $(x_{n+1}, y)$ at $\hat{\theta}$.
7: **Compute ASSLA log-PPD (up to constant):**
8:     $\log \hat{p}_{\text{ASSLA}}(y \mid x_{n+1}, D) \leftarrow \Delta_\ell(y) - \frac{1}{2}\Delta_J(y)$.
9: **return** $\log \hat{p}_{\text{ASSLA}}(y \mid x_{n+1}, D)$.

---

# 4 Approximating Further

From an applied perspective, equation 12 requires the computation of the prior, the log-likelihood, and Fisher information. Specifically, these three functions need to be evaluated both at $\hat{\theta}$ and $\tilde{\theta}$, which means that we have to solve $\tilde{\theta} \in \arg\max_\theta \tilde{\ell}(\theta)$ in addition to the usual $\hat{\theta} \in \arg\max_\theta \ell_D(\theta) = \arg\max_\theta \left[\sum_{i=1}^n \log p(y_i \mid x_i, \theta)\right]$, which can be burdensome for large models.

A natural question, then, is whether the computational overhead given by $\arg\max_\theta \tilde{\ell}(\theta)$ can be reduced. The following results allow us to answer positively to such a query. The challenge is that a simple approximation $\tilde{\theta} \approx \hat{\theta}$ (Lemma 1) does not suffice, since the approximation error thereof might be propagated and increased through $\ell$ and $\tilde{\ell}$. We thus need to derive the approximation error for $\tilde{\ell}(\tilde{\theta}) \approx \tilde{\ell}(\hat{\theta})$. Theorem 1 finds it for Lipschitz-continuous loss function, which implies that we can approximate $\tilde{\ell}(\tilde{\theta}) \approx \tilde{\ell}(\hat{\theta})$ with an approximation error that decreases linearly in $n$.

**Assumptions.** Throughout the following results, we assume:

(A1) The per-sample log-likelihood terms $\log p(y_i \mid x_i, \theta)$ are twice continuously differentiable in a neighborhood of $\hat{\theta}$.

(A2) The observed curvature matrices are positive definite in this neighborhood, with eigenvalues bounded below by a positive constant.

(A3) The log-prior $\log \pi(\theta)$ is twice continuously differentiable with bounded gradient and Hessian in a neighborhood of $\hat{\theta}$.

**Lemma 1.** *Under Assumptions (A1)–(A3), it holds that $\tilde{\theta} = \hat{\theta} + O(n^{-1})$, where $O$ denotes the Bachmann–Landau big-O notation. That is, $\tilde{\theta} \approx \hat{\theta}$ for growing $n$.*

**Theorem 1.** *Assume that the loss is Lipschitz-continuous, and denote by $L$ its Lipschitz constant. Then, under Assumptions (A1)–(A3), we have that*

$$\tilde{\ell}(\tilde{\theta}) = \tilde{\ell}(\hat{\theta}) + O\left(\frac{Ln+1}{n^2}\right).$$

Our approximation in equation 12, however, depends on $\tilde{\theta}$ not only through $\tilde{\ell}(\tilde{\theta})$, but also through the Fisher info $\tilde{\mathcal{J}}(\tilde{\theta})$ and the prior $\pi(\tilde{\theta})$. So we need to take into account the approximation error of $\tilde{\mathcal{J}}(\tilde{\theta}) = -\nabla_\theta^2 \tilde{\ell}(\tilde{\theta}) \approx -\nabla_\theta^2 \tilde{\ell}(\hat{\theta})$ and $\pi(\tilde{\theta}) \approx \pi(\hat{\theta})$. Corollary 1 takes care of the latter two.

**Corollary 1.** *Assume that the likelihood's second derivative $\nabla_\theta^2 \tilde{\ell}$ and the prior $\pi$ are both Lipschitz-continuous. Then, still assuming (A1)–(A3), we have that*

$$\tilde{\mathcal{J}}(\tilde{\theta}) = -\tilde{\ell}''(\tilde{\theta})/n = -\tilde{\ell}''(\hat{\theta})/n + O\left(\frac{Ln+1}{n^2}\right)$$

*and*

$$\pi(\tilde{\theta}) = \pi(\hat{\theta}) + O\left(\frac{Ln+1}{n^2}\right),$$

As we can see, Theorem 1 and Corollary 1 eliminate the need to compute $\tilde{\theta}$ entirely. That is, we do not have to refit (and optimize) the model for both $D$ and $\{(x_{n+1}, \hat{y}_{n+1})\}$. We can restrict ourselves to the parameter vector $\hat{\theta}$, which can be estimated from the initial training of a single non-Bayesian model, e.g., a neural network. We are left to compute the *expanded likelihood* $\tilde{\ell}$ evaluated at $\hat{\theta}$. We also point out how, as a result of the Gauss-Newton approximation (Foresee & Hagan, 1997), the observed Fisher information $\tilde{\mathcal{J}}(\theta) = -\tilde{\ell}''(\theta)/n$ at general $\theta$ can be approximated as

$$\tilde{\mathcal{J}}(\theta) \approx \frac{1}{n} \tilde{G}(\theta)^\top \tilde{G}(\theta), \tag{13}$$

where $\tilde{G}(\theta)$ is the Jacobian vector that we obtain from $\tilde{\ell}$.

**Corollary 2.** *Given Assumptions (A1)–(A3) and Lipschitz-continuous loss, the following is true*

$$\tilde{\mathcal{J}}(\theta) \approx \frac{1}{n} \tilde{G}(\hat{\theta})^\top \tilde{G}(\hat{\theta}).$$

Corollary 2 implies that we only have to compute the Fisher information for $(x_{n+1}, \hat{y}_{n+1})$ with given parameters $\hat{\theta}$. For the example of a BNN, this requires only one pass through the non-Bayesian neural network with weights $\hat{\theta}$.

The posterior predictive distribution under ASSLA therefore becomes:

$$\log(p(\hat{y}_{n+1}|x_{n+1}, D)) \approx \tilde{\ell}(\hat{\theta}) + \frac{1}{2}\log\left(|\mathcal{J}(\hat{\theta})|\right) - \ell(\hat{\theta}) - \frac{1}{2}\log\left(|\tilde{\mathcal{J}}(\hat{\theta})|\right) \tag{14}$$

due to the fact that the difference between $\pi(\tilde{\theta})$ and $\pi(\hat{\theta})$ is of order $O\left(\frac{Ln+1}{n^2}\right)$, see Corollary 1. Recall from this Corollary 1 that the functional form of the prior itself does not depend on $(x_{n+1}, \hat{y}_{n+1})$. Thus, its contribution is subleading relative to the likelihood and curvature increments and can be neglected at the

level of approximation used in ASSLA. In contrast, both the likelihood and the Fisher information change at leading order and therefore remain in the approximation. This formulation bypasses the computational overhead of refitting the model. Due to $\tilde{\ell}(\theta) = \ell_{(x_{n+1}, \hat{y}_{n+1})}(\theta) + \ell(\theta)$, equation 14 becomes

$$\log(p(\hat{y}_{n+1}|x_{n+1}, D)) \approx \ell_{(x_{n+1}, \hat{y}_{n+1})}(\hat{\theta}) + \frac{1}{2}\log\left(|\mathcal{J}(\hat{\theta})|\right) - \frac{1}{2}\log\left(|\tilde{\mathcal{J}}(\hat{\theta})|\right) \tag{15}$$

allowing us to bypass the calculation of the log-likelihood of the training data $\ell(\hat{\theta})$.

**Why the prior increment can be dropped in ASSLA:** Equation 12 contains the prior increment $\log\pi(\tilde{\theta}) - \log\pi(\hat{\theta})$. In ASSLA, we replace the refit optimum $\tilde{\theta}$ by the original optimum $\hat{\theta}$ and keep only leading-order terms in the expansion. By Corollary 1 (and Assumption (A3)), the shift $\Delta = \tilde{\theta} - \hat{\theta}$ is $O(n^{-1})$, and since the prior's functional form is unaffected by $\hat{y}_{n+1}$, the prior increment is of higher order

$$\log\pi(\tilde{\theta}) - \log\pi(\hat{\theta}) = O(\|\Delta\|) = O(n^{-1})$$

or smaller under the stated bounds. In contrast, the likelihood and curvature increments remain leading-order at the approximation order used in ASSLA, and are therefore retained. This explains the apparent disconnect: the *SSLA* decomposition contains the prior increment, while the *ASSLA* simplification drops it as a higher-order term at the chosen approximation order.

## 5 Experiments

We first verify our method in the conjugate case for regression tasks. That is, we use prior distributions which are conjugate to the likelihood and therefore result in a posterior predictive distribution (PPD) that can be computed analytically. We can thus determine whether (A)SSLA provides reliable and close approximations of the true underlying PPDs in a controlled manner, before scaling to more complex models. We compare three types of conjugate prior models: The normal-normal model, the Poisson-gamma model, and a conjugate Bayesian linear regression. All results and detailed descriptions of the experimental setup can be found in appendix B.

To sum up, SSLA and ASSLA are able to recover the analytic PPD close to perfect for sample sizes between $n = 20$ and $n = 100000$. Numerical instabilities emerged in our experiments for $n \geq 1000000$ for ASSLA. The reason for these instabilities primarily lies in the curvature term. Table 1 shows that subtract-two-aggregates computation in float64 yields max $|\log|$-errors on the order of $10^{-11}$–$10^{-10}$ across all $n$, confirming that the observed degradation in float32 is due to finite-precision cancellation rather than a methodological limitation of ASSLA. However, SSLA is still able to match the analytic PPD closely in these cases.[6] For the normal-normal model, Figure 1 illustrates the comparison of the posterior predictive densities for different sample sizes. Similar visualization for the other conjugate cases can be found in appendix B.

After having validated our approximation for the conjugate case, we conduct a twofold benchmark study, consisting of simulated and real world data.

### 5.1 Simulated Heteroscedastic Regression in Neural Networks

Building on the validation in conjugate settings, we next consider a controlled heteroscedastic regression task to evaluate how well uncertainty quantification methods adapt when the noise variance is input-dependent. We compare SSLA and ASSLA to a suite of established approximate Bayesian inference techniques: the classical Laplace approximation (Daxberger et al., 2021a), variational inference including both the mean-field BNN of Blundell et al. (2015) and the more expressive VI model of Depeweg et al. (2018), and Hamiltonian Monte Carlo implemented via `hamiltorch` (Cobb, 2023b) as a sampling-based reference.

---

[6]The transition between the regimes $n \leq 10^5$ and $n \geq 10^6$ is gradual. Additional experiments (see Table 1) in the conjugate normal–normal model for $n \in \{2\cdot10^5, 3\cdot10^5, 5\cdot10^5, 7\cdot10^5, 9\cdot10^5\}$ show that ASSLA remains stable but exhibits increasing numerical error as $n$ grows. The degradation is primarily caused by finite-precision cancellation: ASSLA computes predictive increments by subtracting two large aggregated log-likelihood terms in `float32`, which can collapse small differences to zero. Repeating the same computation in `float64` eliminates this effect, confirming that the instability is numerical rather than methodological.

**Comparison of SSLA and Analytic Posterior Predictive Distributions**

Figure 1: Conjugate normal-normal model: Six comparisons of SSLA (red) and ASSLA (black) to analytic posterior predictive distribution (green) for varying sample sizes $n$ ranging from $n = 20$ to $n = 1000000$. SSLA and ASSLA are able to closely match the analytic distributions, highlighted by low entropy scores.

The synthetic data are generated with the `ToyHeteroscedasticDataModule` from Lightning-UQ-Box (Lehmann et al., 2024). Inputs $x$ are drawn from a mixture of three Gaussians (centers at $x_{\min} = -4$, 0, and $x_{\max} = 4$ with standard deviations 0.4, 0.9, and 0.4, respectively), and targets follow

$$y = 7\sin(x) + \epsilon, \qquad \epsilon = 3\left|\cos\left(\frac{x}{2}\right)\right| \cdot \mathcal{N}(0, 1), \tag{16}$$

so that the noise amplitude varies smoothly with $x$. A multilayer perceptron (Bishop & Nasrabadi, 2006; Peng, 2017) is used as the predictive model; architectural choices, loss definitions, and hyperparameter selection are detailed in Appendix C. For the Laplace-based methods (including SSLA and ASSLA), the observed Fisher information is approximated via three covariance representations: diagonal, Kronecker-factored (KFAC), and dense. We additionally report Lightning UQ Box (Lehmann et al., 2024) results in Appendix D.

Figure 2 depicts the posterior predictive means with associated credible bands, and Table 2 reports empirical coverage at nominal confidence levels of 95%, 90%, 75%, and 50%. The classical Laplace approximation exhibits conservative calibration, producing wide, risk-averse intervals. SSLA's uncertainty estimates vary depending on the covariance approximation and suffer from instability in calibration.

ASSLA yields smoother and comparatively tighter credible regions, particularly in narrower intervals, but systematically underestimates uncertainty in wider intervals, leading to a mild risk-seeking bias. This behavior reflects the core approximation in ASSLA: replacing the augmented optimum $\tilde{\theta}$ by the original fit $\hat{\theta}$ neglects refitting-induced curvature changes. When the pseudo-observation would have shifted the optimum non-negligibly, ASSLA produces overly concentrated predictive densities. Consequently, ASSLA balances computational tractability with competitive calibration in moderate regimes, but may under-cover at high nominal levels in settings where refitting effects are substantial. In such regimes, SSLA—or more conservative approximations—should be preferred.

| $n$ | runs | KL (mean $\pm$ sd) | Max $|\log|$ err (mean $\pm$ sd) | Frac($\delta$=0) | Frac(nonfinite) |
|---|---|---|---|---|---|
| 100,000 | 6 | $9.41{\times}10^{-6} \pm 1.3{\times}10^{-6}$ | $7.57{\times}10^{-3} \pm 3.1{\times}10^{-4}$ | 0.00 | 0.00 |
| 200,000 | 6 | $5.02{\times}10^{-5} \pm 2.3{\times}10^{-6}$ | $1.55{\times}10^{-2} \pm 2.7{\times}10^{-5}$ | 0.00 | 0.00 |
| 300,000 | 6 | $1.41{\times}10^{-4} \pm 3.3{\times}10^{-6}$ | $3.05{\times}10^{-2} \pm 5.1{\times}10^{-4}$ | 0.00 | 0.00 |
| 500,000 | 6 | $1.40{\times}10^{-4} \pm 3.3{\times}10^{-6}$ | $3.06{\times}10^{-2} \pm 3.1{\times}10^{-4}$ | 0.00 | 0.00 |
| 700,000 | 6 | $4.75{\times}10^{-4} \pm 4.6{\times}10^{-7}$ | $5.86{\times}10^{-2} \pm 2.1{\times}10^{-3}$ | 0.00 | 0.00 |
| 900,000 | 6 | $4.76{\times}10^{-4} \pm 2.0{\times}10^{-6}$ | $5.72{\times}10^{-2} \pm 2.7{\times}10^{-3}$ | 0.00 | 0.00 |
| 1,000,000 | 6 | $4.75{\times}10^{-4} \pm 9.2{\times}10^{-7}$ | $5.69{\times}10^{-2} \pm 2.6{\times}10^{-3}$ | 0.00 | 0.00 |

Table 1: Intermediate regime $n \in [10^5, 10^6]$ in the Normal–Normal conjugate setting: the transition is gradual rather than abrupt. Reported numbers compare a numerically stable reference computation against an ASSLA float32 computation, which illustrates finite-precision cancellation effects.

Figure 2: We display the predictive uncertainty intervals for various uncertainty quantification methods, namely SSLA, ASSLA, LA, VI, and MCMC. The shaded blue areas represent the 95% credible intervals, with dots representing observations and the solid blue line indicating the model's predictions.

The flexible variational inference model tracks nominal coverage most closely with minimal systematic deviation, whereas MFVI collapses toward the predictive mean and manifests risk-averse behavior. HMC, despite being a theoretically grounded baseline, reflects practical linearization limitations in this implementation and delivers intermediate coverage (e.g., approximately 76% at the 75% level and 68% at 50%), capturing the underlying structure without extreme dispersion.

These quantitative and qualitative findings expose a nuanced trade-off surface: SSLA can approximate coverage in certain regimes but is hampered by stability concerns; ASSLA strikes a favorable balance in computational efficiency and interval smoothness relative to classical Laplace and HMC, yet its broader underestimation in wide intervals requires caution; variational inference provides the most balanced empirical calibration; and the standard Laplace method remains the most conservative. Together with the earlier conjugate-case insights, this comparison refines practical guidance for selecting posterior-predictive approximation techniques under heteroscedastic neural regression. (Staber & Veiga, 2023)

| CI | SSLA | | | ASSLA | | | LA | | | VI | | MCMC |
|---|---|---|---|---|---|---|---|---|---|---|---|---|
| | KFAC | DIAG | DENSE | KFAC | DIAG | DENSE | KFAC | DIAG | DENSE | VI | MFVI | HMC |
| 95% | 88.0 | 92.0 | 92.0 | 76.0 | 76.0 | 76.0 | 100.0 | 100.0 | 100.0 | 92.0 | 100.0 | 88.0 |
| 90% | 84.0 | 92.0 | 88.0 | 68.0 | 68.0 | 68.0 | 100.0 | 100.0 | 100.0 | 92.0 | 100.0 | 88.0 |
| 75% | 76.0 | 80.0 | 76.0 | 60.0 | 60.0 | 60.0 | 92.0 | 92.0 | 92.0 | 80.0 | 96.0 | 76.0 |
| 50% | 72.0 | 64.0 | 56.0 | 56.0 | 56.0 | 56.0 | 68.0 | 68.0 | 68.0 | 64.0 | 80.0 | 68.0 |

Table 2: We report the coverage of different methods at various confidence intervals for heteroscedastic regression. The table shows the coverage percentages for SSLA, ASSLA, LA, VI, and MCMC methods, evaluated using KFAC, DIAG, and DENSE for each approach at the 95%, 90%, 75%, and 50% confidence intervals.

| Dataset | Description | Size | Subsampling ($n = 50$) |
|---|---|---|---|
| Concrete Compressive Strength (Yeh, 1998) | Modeling the compressive strength of concrete using 8 features. | Medium | ✓ |
| Wine Quality (Cortez & Reis, 2009) | Portuguese "Vinho Verde" wine, with quality scores ranging from 0 to 10. | Medium | ✓ |
| Bike Sharing (Fanaee-T, 2013) | Prediction of total rental bike counts using hourly and daily data. | Medium | ✓ |
| Airfoil Self-Noise (Brooks & Marcolini, 1989) | NASA aerodynamic and acoustic test results for airfoil blade sections (Moin et al., 2022). | Medium | ✓ |
| Auto MPG (Quinlan, 1993) | Predicting the "mpg" attribute using 7 features. | Small | ✗ |
| Liver Disorders (Asuncion et al., 2016) | Blood test results used to predict daily alcoholic beverage consumption. | Small | ✗ |
| Daily Demand Forecasting (Ferreira & Sassi, 2016) | Daily demand forecasting in a Brazilian logistics company. | Small | ✗ |
| Real Estate Valuation (Yeh, 2018) | Historical market data for real estate valuation in Taiwan. | Small | ✗ |

Table 3: Overview of datasets used for benchmarking ASSLA and SSLA from UCIMLRepo (Kelly et al., 2025). Subsampling is applied to medium-sized datasets.

## 5.2 Case Study on Real-World Data

To assess the empirical viability of (A)SSLA beyond controlled synthetic settings, we apply the methods to a diverse set of real-world regression tasks from the UCI Machine Learning Repository (Kelly et al., 2025). These datasets span domains and scales, providing a practical stress test for uncertainty quantification under realistic data artifacts and varying sample regimes. Table 3 summarizes the datasets, their modifications, and the subsampling strategy; SSLA is restricted to small datasets (fewer than 500 observations) due to its computational overhead, whereas ASSLA is employed more broadly with subsampling on medium-sized data to maintain tractability.

All datasets are processed through a unified pipeline: binary features are encoded as 0/1, categorical variables are one-hot expanded, and continuous inputs and targets are standardized to zero mean and unit variance. Several dataset-specific adaptations reframe the raw data into regression tasks or emphasize robustness: the Wine Quality data incorporates the color attribute and predicts alcohol content rather than quality; Bike Sharing discards the datetime column and predicts normalized ambient temperature; Daily Demand Forecasting Orders is treated as a regression problem via random sampling of train/test splits; Liver Disorders predicts alkaline phosphatase (ALP) instead of alcohol intake, reflecting a biomedically relevant proxy for liver/bone pathology (Lowe et al., 2025); and Auto MPG omits the non-informative `car_names` field.

| Dataset | Method | Coverage | | | NLL ↓ | CRPS ↓ |
|---|---|---|---|---|---|---|
| | | 95% | 75% | 50% | | |
| Auto MPG | SSLA | 100.00 | 91.14 | 74.68 | 0.45 | 0.19 |
| | ASSLA | 100.00 | 82.28 | 69.62 | 0.32 | 0.18 |
| | LA | 100.00 | 100.00 | 94.94 | 0.99 | 0.28 |
| Liver Disorders | SSLA | 73.91 | 44.93 | 28.99 | 2.74 | 0.79 |
| | ASSLA | 49.28 | 24.64 | 15.94 | 5.94 | 0.85 |
| | LA | 91.30 | 66.67 | 43.48 | 1.68 | 0.73 |
| Concrete Compressive Strength | SSLA | 98.00 | 98.00 | 82.00 | 0.43 | 0.19 |
| | ASSLA | 98.00 | 90.00 | 72.00 | 0.38 | 0.18 |
| | LA | 100.00 | 98.00 | 98.00 | 1.00 | 0.28 |
| Wine Quality | SSLA | 98.00 | 88.00 | 72.00 | 0.49 | 0.21 |
| | ASSLA | 94.00 | 84.00 | 62.00 | 0.39 | 0.20 |
| | LA | 100.00 | 100.00 | 90.00 | 0.99 | 0.28 |
| Bike Sharing | SSLA | 100.00 | 100.00 | 100.00 | 0.13 | 0.11 |
| | ASSLA | 100.00 | 100.00 | 100.00 | 0.00 | 0.09 |
| | LA | 100.00 | 100.00 | 100.00 | 0.92 | 0.23 |
| Airfoil Self-Noise | SSLA | 98.00 | 94.00 | 88.00 | 0.51 | 0.19 |
| | ASSLA | 96.00 | 90.00 | 84.00 | 0.24 | 0.16 |
| | LA | 100.00 | 100.00 | 94.00 | 0.97 | 0.27 |
| Daily Demand Forecasting Orders | SSLA | 91.67 | 91.67 | 91.67 | 0.44 | 0.18 |
| | ASSLA | 91.67 | 91.67 | 83.33 | 0.34 | 0.17 |
| | LA | 100.00 | 100.00 | 91.67 | 1.06 | 0.29 |
| Real Estate Valuation | SSLA | 97.59 | 93.98 | 79.52 | 0.96 | 0.32 |
| | ASSLA | 90.36 | 74.70 | 50.60 | 0.74 | 0.26 |
| | LA | 100.00 | 97.59 | 86.75 | 1.05 | 0.32 |

Table 4: Coverage, NLL, and CRPS results for SSLA, ASSLA, and LA on various UCIMLRepo datasets

The predictive model mirrors the heteroscedastic setting: a multilayer perceptron with two hidden layers of 50 units each and ReLU nonlinearities is trained to represent both mean and uncertainty. Hyperparameters are selected via Optuna, searching learning rates in $[1 \times 10^{-5}, 1 \times 10^{-1}]$ and batch sizes from 32 to 256 over 100 trials, with validation negative log-likelihood governing selection. For SSLA we assume a standard normal prior on weights, apply last-layer Laplace approximation with post-hoc tuning as in Daxberger et al. (2021a, Chapter 4.1), and approximate the observed Fisher information using KFAC. All remaining training and loss configurations are consistent with the heteroscedastic regression experiment.

Table 4 reports empirical coverage at nominal 95%, 75%, and 50% credible intervals alongside negative log-likelihood (NLL) and continuous ranked probability score (CRPS), offering both calibration and sharpness diagnostics. The methods exhibit dataset-dependent trade-offs. In Auto MPG both SSLA and ASSLA achieve full coverage at 95%, but ASSLA attains substantially lower NLL than SSLA and the classical Laplace approximation, indicating tighter yet well-calibrated uncertainty. The Liver Disorders task exposes a failure mode: both SSLA and ASSLA underperform relative to standard Laplace, with ASSLA showing pronounced undercoverage and inflated NLL, suggesting its adjustment mechanism can over-compress uncertainty when the signal-to-noise ratio is poor.

On Concrete Compressive Strength and Wine Quality, SSLA and ASSLA reach near-nominal high-level coverage while delivering considerably better NLL and CRPS than the overly conservative Laplace baseline, which produces wide intervals that dilute practical informativeness. The Bike Sharing dataset is a strong success case for ASSLA: all methods achieve perfect coverage, yet ASSLA yields near-zero NLL and the lowest CRPS, demonstrating its ability to produce highly concentrated, calibrated predictive distributions in favorable signal regimes. For Airfoil Self-Noise and Daily Demand Forecasting Orders, ASSLA consistently improves upon SSLA in efficiency (lower NLL) with comparable coverage, reinforcing its robustness in medium-scale real-world data. In Real Estate Valuation, SSLA slightly edges ASSLA in coverage at 95%, but ASSLA achieves a more favorable balance between uncertainty calibration and predictive quality than the conservative Laplace approach, whose inflated uncertainty (and higher NLL/CRPS) reduces sharpness.

These empirical results show a nuanced performance landscape. This kind of multi-criteria evaluation naturally leads to comparisons that need not induce a total order (Jansen et al., 2024a;b; Rodemann & Blocher, 2024; Arias et al., 2025a;b); we thus discuss the arising trade-offs in what follows. ASSLA frequently delivers sharper and better-calibrated uncertainty estimates than classical Laplace, particularly where the latter's risk-aversion would degrade downstream utility. SSLA can sometimes yield marginally higher coverage but incurs higher computational cost and may suffer instability in more challenging regimes. The degradation on Liver Disorders highlights that ASSLA's adjustment may overfit uncertainty compression when evidence is weak, indicating a regime where fallback to more conservative approximations or hybrid regularization might be necessary. Overall, the case study corroborates ASSLA as a practically appealing method for real-world regression: it balances computational tractability with competitive probabilistic calibration across diverse settings while signaling scenarios requiring caution (Staber & Veiga, 2023).

## 6 Limitations and Conclusion

In this work, we proposed to shift the focus of approximate Bayesian inference from the intractable parameter posterior to the posterior predictive distribution, which is the quantity of primary practical interest for uncertainty-aware prediction. Our Self-Supervised Laplace Approximation (SSLA) quantifies predictive uncertainty by refitting on model-generated (self-predicted) data: predictions that the model assigns high likelihood are identified with low uncertainty, and vice versa. This self-supervised mechanism is modular in the prior, enabling sensitivity analysis across prior choices. To reduce the computational burden of refitting, we derived an approximate variant (ASSLA) that leverages asymptotic expansions and local linearization to express the posterior predictive in terms of quantities evaluated at the original fit, avoiding expensive re-optimization.

Theoretical results characterize the approximation error and justify replacing the augmented posterior mode with the original mode under mild regularity, controlling deviations in the likelihood, Fisher information, and prior. Empirically, we benchmark SSLA and ASSLA across a hierarchy of settings: from conjugate analytic models—where ground-truth posterior predictives are available—to controlled synthetic heteroscedastic regression and a diverse suite of real-world regression tasks. Comparisons include classical Laplace, variational inference (including mean-field and more expressive variants), and Hamiltonian Monte Carlo, evaluating both calibration (coverage) and sharpness. In controlled settings such as conjugate prior scenarios, both SSLA and ASSLA closely approximate the analytical posterior predictive distribution. In more complex environments like heteroscedastic regression and real-world datasets, ASSLA generally delivers smoother and more computationally efficient uncertainty estimates than SSLA, while traditional methods such as LA often exhibit overly conservative behavior.

However, several challenges remain. SSLA can become unstable on larger datasets or in the presence of prior-data conflicts (Evans & Moshonov, 2006; Walter & Augustin, 2009; Marquardt et al., 2023), and ASSLA sometimes underestimates uncertainty in regions with high variance. These findings indicate that while (A)SSLA are promising, they require further refinement to handle all scenarios robustly. First, refining the covariance approximation and addressing numerical instabilities could substantially enhance the reliability of these methods. In some scenarios, it may even be possible to circumvent the full computation of the FIM or to approximate it more efficiently, potentially yielding significant performance gains. Second, given the prior modularity in SSLA, it would be interesting to explore its application within the framework of imprecise probability. In this setting, one could leverage extreme points of prior credal sets to derive an imprecise posterior predictive distribution.

Together, these studies demonstrate that ASSLA in particular often achieves a favorable trade-off between computational efficiency and predictive uncertainty quality, producing sharper and better-calibrated predictive distributions than standard Laplace in many regimes, while SSLA provides a more faithful (but costlier) self-refitting baseline.

More generally SSLA/ASSLA do not guarantee that the model likelihood is reliable under arbitrary misspecification or severe dataset shift: like any Bayesian or likelihood-based uncertainty quantification method, they inherit the assumptions of the chosen likelihood model. Conditional on the likelihood being a meaningful local model around $\hat{\theta}$, SSLA/ASSLA quantify uncertainty primarily through *changes* induced by the

pseudo-observation—objective and curvature increments—rather than relying on the absolute log-likelihood value alone. Under drift, these increments can still surface atypical curvature behavior (e.g., poor conditioning and sensitive log-determinants), but any likelihood-based score must be interpreted with caution. In practice, SSLA/ASSLA should be used alongside drift/OOD detection and, where appropriate, robust likelihood modeling or likelihood tempering.

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

## A  Proofs

*Proof of Lemma 1.* By definition, $\hat{\theta}$ and $\tilde{\theta}$ satisfy

$$\nabla \ell_D(\hat{\theta}) + \nabla \log \pi(\hat{\theta}) = 0, \qquad \nabla \tilde{\ell}(\tilde{\theta}) + \nabla \log \pi(\tilde{\theta}) = 0,$$

where $\tilde{\ell}(\theta) = \ell_D(\theta) + \log p(\hat{y}_{n+1} \mid x_{n+1}, \theta)$. Subtracting the two equations and applying a first-order Taylor expansion of the gradients around $\hat{\theta}$ yields

$$0 = \nabla \tilde{\ell}(\tilde{\theta}) + \nabla \log \pi(\tilde{\theta}) = \nabla \tilde{\ell}(\hat{\theta}) + \nabla \log \pi(\hat{\theta}) + \tilde{J}(\bar{\theta})(\tilde{\theta} - \hat{\theta}),$$

for some $\bar{\theta}$ on the line segment between $\hat{\theta}$ and $\tilde{\theta}$.

By Assumptions (A1)–(A3), $\tilde{J}(\bar{\theta})$ is positive definite with eigenvalues bounded below, hence invertible with uniformly bounded inverse. Moreover,

$$\nabla \tilde{\ell}(\hat{\theta}) + \nabla \log \pi(\hat{\theta}) = \nabla \ell_D(\hat{\theta}) + \nabla \log \pi(\hat{\theta}) + \nabla \log p(\hat{y}_{n+1} \mid x_{n+1}, \hat{\theta}) = \nabla \log p(\hat{y}_{n+1} \mid x_{n+1}, \hat{\theta}),$$

which is $O(1)$.

Since $\tilde{J}(\bar{\theta}) = O(n)$, it follows that

$$\tilde{\theta} - \hat{\theta} = -\tilde{J}(\bar{\theta})^{-1} \nabla \log p(\hat{y}_{n+1} \mid x_{n+1}, \hat{\theta}) = O(n^{-1}),$$

which proves the claim. $\square$

*Proof of Theorem 1.* We work under assumptions (A1)–(A3):

(A1) $\ell_D(\theta) = \sum_{i=1}^n \ell_i(\theta)$ with each $\ell_i$ twice continuously differentiable in a neighborhood of $\hat{\theta}$.

(A2) The observed curvature $\mathcal{J}_D(\hat{\theta})$ is positive definite with eigenvalues bounded below by $\lambda_{\min} > 0$.

(A3) The prior log-density $\log \pi(\theta)$ is twice continuously differentiable with bounded gradient and Hessian in a neighborhood of $\hat{\theta}$.

In addition, we use the Lipschitz condition on the single-point term stated in Theorem 1.

First, notice that $\tilde{\ell}$ *contains* $\ell$, that is,

$$\tilde{\ell}(\theta) := \ell_{(x_{n+1}, \hat{y}_{n+1})}(\theta) + \ell(\theta).$$

In order to simplify the computation of $\tilde{\ell}(\tilde{\theta})$, we expand $\ell$ around its maximizer $\hat{\theta}$, so that

$$\ell(\tilde{\theta}) = \ell(\hat{\theta}) + O(\|\hat{\theta} - \tilde{\theta}\|_2^2) \tag{17}$$

By Lemma 1, we know that since $D \cup \{(x_{n+1}, \hat{y}_{n+1})\}$ and $D$ differ in only one sample, the difference $\hat{\theta} - \tilde{\theta}$ is of order $O(n^{-1})$. Combining this with equation 17, we have

$$\tilde{\ell}(\tilde{\theta}) = \ell_{(x_{n+1}, \hat{y}_{n+1})}(\tilde{\theta}) + \ell(\hat{\theta}) + O(n^{-2}),$$

which entails

$$\tilde{\ell}(\tilde{\theta}) \approx \ell_{(x_{n+1}, \hat{y}_{n+1})}(\tilde{\theta}) + \ell(\hat{\theta}). \tag{18}$$

We then ask ourselves if $\ell_{(x_{n+1}, \hat{y}_{n+1})}(\tilde{\theta})$ can be approximated by $\ell_{(x_{n+1}, \hat{y}_{n+1})}(\hat{\theta})$. Recall that the difference $\hat{\theta} - \tilde{\theta}$ is of order $O(n^{-1})$, as $D \cup \{(x_{n+1}, \hat{y}_{n+1})\}$ and $D$ differ in only one sample. In addition, the likelihood function $\ell_{(x_{n+1}, \hat{y}_{n+1})}(\cdot)$ is Lipschitz-continuous in $\theta$. This holds per assumption of Lipschitz-continuous loss.

Now consider the Lipschitz bound

$$\left\| \ell_{(x_{n+1}, \hat{y}_{n+1})}(\tilde{\theta}) - \ell_{(x_{n+1}, \hat{y}_{n+1})}(\hat{\theta}) \right\|_2 \leq L \cdot \|\hat{\theta} - \tilde{\theta}\|_2, \tag{19}$$

where $L$ is a Lipschitz constant. Exploiting the fact that $\hat{\theta} - \tilde{\theta}$ is of order $O(n^{-1})$, Equation equation 19 entails that

$$\ell_{(x_{n+1}, \hat{y}_{n+1})}(\tilde{\theta}) = \ell_{(x_{n+1}, \hat{y}_{n+1})}(\hat{\theta}) \pm O\left(\frac{L}{n}\right), \tag{20}$$

where the remainder is negligible as $n \to \infty$. Hence, equation 20 implies that

$$\ell_{(x_{n+1}, \hat{y}_{n+1})}(\tilde{\theta}) \approx \ell_{(x_{n+1}, \hat{y}_{n+1})}(\hat{\theta}). \tag{21}$$

Combining equation 21 with equation 18, we have

$$\tilde{\ell}(\tilde{\theta}) \approx \ell_{(x_{n+1}, \hat{y}_{n+1})}(\hat{\theta}) + \ell(\hat{\theta}) = \tilde{\ell}(\hat{\theta}) \tag{22}$$

and, as a by-product,

$$\ell(\tilde{\theta}) \approx \ell(\hat{\theta}).$$

We can express Equation (22), $\tilde{\ell}(\tilde{\theta}) \approx \tilde{\ell}(\hat{\theta})$, equivalently in asymptotic (Bachmann–Landau) notation

$$\tilde{\ell}(\tilde{\theta}) = \tilde{\ell}(\hat{\theta}) + O\left(\frac{L}{n}\right) + O(n^{-2}) = \tilde{\ell}(\hat{\theta}) + O\left(\frac{Ln + 1}{n^2}\right)$$

by combining equations (20) and (18).

$\square$

*Proof of Corollary 1.* This corollary is proved under the same regularity conditions as Theorem 1, namely (A1)–(A3). Recall from the proof of Theorem 1 that $\tilde{\ell}$ contains $\ell_D$. Thus the same holds for their derivatives if they exist. Further recall that $D \cup \{(x_{n+1}, \hat{y}_{n+1})\}$ and $D$ differ in only one sample. So the difference $\hat{\theta} - \tilde{\theta}$ is of order $O(n^{-1})$. By requiring Lipschitz-continuity for the second derivatives and the prior, we can apply the same reasoning as in the proof of Theorem 1. Hence, it follows that

$$-\tilde{\ell}''(\tilde{\theta})/n = -\tilde{\ell}''(\hat{\theta})/n + O\left(\frac{Ln + 1}{n^2}\right)$$

and

$$\pi(\tilde{\theta}) = \pi(\hat{\theta}) + O\left(\frac{Ln + 1}{n^2}\right),$$

or simply

$$\mathcal{I}(\tilde{\theta}) = -\tilde{\ell}''(\tilde{\theta})/n \approx -\tilde{\ell}''(\hat{\theta})/n \quad \text{and} \quad \pi(\tilde{\theta}) \approx \pi(\hat{\theta}).$$

□

*Proof of Corollary 2.* Immediate from Theorem 1 and Corollary 1. □

### A.1  Highest Density Region Approximation

Notice that the integral $\int_\Theta \exp\left[-\frac{n}{2}(\theta - \tilde{\theta})^\top \mathcal{I}(\tilde{\theta})(\theta - \tilde{\theta})\right]\mathrm{d}\theta$ in equation 7 is a Gaussian integral, and so $\theta$ follows a multivariate normal distribution (Kass et al., 1990). In addition, it is well-known that any marginal distribution of a multivariate normal distribution is again a multivariate normal. As a consequence, our approximation of $p(\hat{y}_{n+1} \mid x_{n+1}, D)$ is a multivariate normal and thus symmetric around its mode. This implies $R(\hat{p}^\alpha)$ is symmetric around a given $\hat{y} \in \mathcal{Y}$, i.e., $R(\hat{p}^\alpha) = [\hat{y} - b, \hat{y} + b]$. This simplifies the solution for $\hat{p}^\alpha$. We have

$$P[\hat{Y}_{n+1} \in R(\hat{p}^\alpha) \mid x_{n+1}, D] \geq 1 - \alpha \iff \int_{R(\hat{p}^\alpha)} p(\hat{y}_{n+1} \mid x_{n+1}, D)\mathrm{d}y \geq 1 - \alpha$$

$$\iff \int_{\hat{y}-b}^{\hat{y}+b} p(\hat{y}_{n+1} \mid x_{n+1}, D)\mathrm{d}y \geq 1 - \alpha.$$

Approximating $p(\hat{y}_{n+1} \mid x_{n+1}, D)$ by Equation equation 12, we obtain

$$\int_{\hat{y}-b}^{\hat{y}+b} \exp\left[\frac{q}{2}\log\left(\frac{n}{n+1}\right) + \tilde{\ell}(\hat{\theta}) - \frac{1}{2}\log|\mathcal{I}(\hat{\theta})| + \frac{1}{2}\log|\mathcal{I}_{\ell_D}(\hat{\theta})| - \ell_D(\hat{\theta}) + \log\pi(\hat{\theta}) - \log\pi(\hat{\theta})\right]\mathrm{d}y$$

$$\geq 1 - \alpha$$

$$\iff \int_{\hat{y}-b}^{\hat{y}+b} \exp\tilde{\ell}(\hat{\theta})\mathrm{d}y$$

$$\geq (1-\alpha)\exp\left[-\frac{q}{2}\log\left(\frac{n}{n+1}\right) + \frac{1}{2}\log|\mathcal{I}(\hat{\theta})| - \frac{1}{2}\log|\mathcal{I}_{\ell_D}(\hat{\theta})| + \ell_D(\hat{\theta}) + \log\pi(\hat{\theta}) - \log\pi(\hat{\theta})\right]$$

$$\iff \int_{\hat{y}-b}^{\hat{y}+b} \mathcal{L}_{\tilde{y},\tilde{x}}(\hat{\theta}, y, x)\mathrm{d}y$$

$$\geq (1-\alpha)\exp\left[-\frac{q}{2}\log\left(\frac{n}{n+1}\right) + \frac{1}{2}\log|\mathcal{I}(\hat{\theta})| - \frac{1}{2}\log|\mathcal{I}_{\ell_D}(\hat{\theta})| + \ell_D(\hat{\theta}) + \log\pi(\hat{\theta}) - \log\pi(\hat{\theta})\right], \quad (23)$$

$$\int_{\hat{y}-b}^{\hat{y}+b} \exp\left[\frac{3q}{2}\log\left(\frac{2\pi}{n}\right) + \tilde{\ell}(\hat{\theta}) - \frac{1}{2}\log|\mathcal{I}(\hat{\theta})| - \log|\mathcal{I}_{\ell_D}(\hat{\theta})| + 2\ell_D(\hat{\theta}) + 3\log\pi(\hat{\theta})\right] dy$$

$$\geq 1 - \alpha$$

$$\iff \int_{\hat{y}-b}^{\hat{y}+b} \exp\tilde{\ell}(\hat{\theta})dy$$

$$\geq (1-\alpha)\exp\left[-\frac{3q}{2}\log\left(\frac{2\pi}{n}\right) + \frac{1}{2}\log|\mathcal{I}(\hat{\theta})| + \log|\mathcal{I}_{\ell_D}(\hat{\theta})| - 2\ell_D(\hat{\theta}) - 3\log\pi(\hat{\theta})\right]$$

$$\iff \int_{\hat{y}-b}^{\hat{y}+b} \mathcal{L}_{\tilde{y},\tilde{x}}(\hat{\theta},y,x)dy$$

$$\geq (1-\alpha)\exp\left[-\frac{3q}{2}\log\left(\frac{2\pi}{n}\right) + \frac{1}{2}\log|\mathcal{I}(\hat{\theta})| + \log|\mathcal{I}_{\ell_D}(\hat{\theta})| - 2\ell_D(\hat{\theta}) - 3\log\pi(\hat{\theta})\right], \tag{24}$$

where $\tilde{y} := (y_1, \ldots, y_n, \hat{y}_{n+1})$ and $\tilde{x} := (x_1, \ldots, x_n, \hat{x}_{n+1})$. Note that per Lemma 1 we have $\log\pi(\hat{\theta}) - \log\pi(\hat{\theta}) = 0$ .

Recall that we selected the $L^2$-loss. Then, we have that

$$\int_{\hat{y}-b}^{\hat{y}+b} \mathcal{L}(\hat{\theta}, y, x)dy = \int_{\hat{y}-b}^{\hat{y}+b} (y - f(\hat{\theta}, x))^2 dy$$

$$= \frac{1}{3}(\hat{y} + b - f(x,\hat{\theta}))^3 - \frac{1}{3}(\hat{y} - b - f(x,\hat{\theta}))^3$$

$$= \frac{1}{3}b^3 - \frac{1}{3}(-b)^3 = \frac{2}{3}b^3. \tag{25}$$

Plugging equation 25 into equation 24, we get that $b$ is lower bounded by

$$\sqrt[3]{\frac{3}{2}(1-\alpha)\exp\left[-\frac{3q}{2}\log\left(\frac{2\pi}{n}\right) - \tilde{\ell}(\hat{\theta}) + \frac{1}{2}\log|\mathcal{I}(\hat{\theta})| + \log|\mathcal{I}_{\ell_D}(\hat{\theta})| - 2\ell_D(\hat{\theta}) - 3\log\pi(\hat{\theta})\right]}. \tag{26}$$

Since we are looking for the smallest possible region $R(\hat{p}^\alpha)$, we want the smallest possible value of $b$. Such smallest possible value is exactly the lower bound in equation 26, which we denote by $b^\star$.

We can now derive $\hat{p}^\alpha$,

$$\hat{p}^\alpha = p(\hat{y}_{n+1} = \hat{y} + b^\star \mid x_{n+1}, D)$$

$$= \left|\hat{y} + b^\star - f(\hat{\theta}, x)\right| \exp\left[\frac{3q}{2}\log\left(\frac{2\pi}{n}\right) - \frac{1}{2}\log|\mathcal{I}(\hat{\theta})| - \log|\mathcal{I}_{\ell_D}(\hat{\theta})| + 2\ell_D(\hat{\theta}) + 3\log\pi(\hat{\theta})\right]$$

$$= |b^\star| \exp\left[\frac{3q}{2}\log\left(\frac{2\pi}{n}\right) - \frac{1}{2}\log|\mathcal{I}(\hat{\theta})| - \log|\mathcal{I}_{\ell_D}(\hat{\theta})| + 2\ell_D(\hat{\theta}) + 3\log\pi(\hat{\theta})\right]$$

$$= \left|\sqrt[3]{\frac{3}{2}(1-\alpha)\exp\left[-\frac{3q}{2}\log\left(\frac{2\pi}{n}\right) + \frac{1}{2}\log|\mathcal{I}(\hat{\theta})| + \log|\mathcal{I}_{\ell_D}(\hat{\theta})| - 2\ell_D(\hat{\theta}) - 3\log\pi(\hat{\theta})\right]}\right|$$

$$\cdot \exp\left[\frac{3q}{2}\log\left(\frac{2\pi}{n}\right) - \frac{1}{2}\log|\mathcal{I}(\hat{\theta})| - \log|\mathcal{I}_{\ell_D}(\hat{\theta})| + 2\ell_D(\hat{\theta}) + 3\log\pi(\hat{\theta})\right]$$

$$= \left|\sqrt[3]{\frac{3}{2}(1-\alpha)}\exp\left[-\frac{q}{2}\log\left(\frac{2\pi}{n}\right) + \frac{1}{6}\log|\mathcal{I}(\hat{\theta})| + \frac{1}{6}\log|\mathcal{I}_{\ell_D}(\hat{\theta})| - \frac{2}{3}\ell_D(\hat{\theta}) - \log\pi(\hat{\theta})\right]\right|$$

$$\cdot \exp\left[\frac{3q}{2}\log\left(\frac{2\pi}{n}\right) - \frac{1}{2}\log|\mathcal{I}(\hat{\theta})| - \log|\mathcal{I}_{\ell_D}(\hat{\theta})| + 2\ell_D(\hat{\theta}) + 3\log\pi(\hat{\theta})\right]$$

$$= \left|\sqrt[3]{\frac{3}{2}(1-\alpha)}\right|\exp\left[-q\log\left(\frac{2\pi}{n}\right) + \frac{1}{3}\log|\mathcal{I}(\hat{\theta})| + \frac{1}{3}\log|\mathcal{I}_{\ell_D}(\hat{\theta})| - \frac{4}{3}\ell_D(\hat{\theta}) - 2\log\pi(\hat{\theta})\right]$$

$$\cdot \exp\left[\frac{3q}{2}\log\left(\frac{2\pi}{n}\right) - \frac{1}{2}\log|\mathcal{I}(\hat{\theta})| - \log|\mathcal{I}_{\ell_D}(\hat{\theta})| + 2\ell_D(\hat{\theta}) + 3\log\pi(\hat{\theta})\right]$$

$$= \sqrt[3]{\frac{9}{4}}(1-\alpha)^{\frac{2}{3}}\exp\left[-\frac{q}{2}\log\left(\frac{2\pi}{n}\right) - \frac{1}{6}\log|\mathcal{I}(\hat{\theta})| - \frac{2}{3}\log|\mathcal{I}_{\ell_D}(\hat{\theta})| + \frac{2}{3}\ell_D(\hat{\theta}) + \log\pi(\hat{\theta})\right].$$

Now that we found $\hat{p}^\alpha$, we focus on finding $\hat{p}^\alpha_{\mathcal{L}}$. We have

$$\log p(\hat{y}_{n+1} \mid x_{n+1}, D) \geq \log(\hat{p}^\alpha) \iff$$

$$\tilde{\ell}(\tilde{\theta}) \geq \log(\hat{p}^\alpha) - \frac{3q}{2}\log\left(\frac{2\pi}{n}\right) - \log\pi(\tilde{\theta}) + \frac{1}{2}\log|\mathcal{I}(\tilde{\theta})|$$

$$+ \log|\mathcal{I}_{\ell_D}(\hat{\theta})| - 2\ell_D(\hat{\theta}) - 2\log\pi(\hat{\theta}) \iff$$

$$\log\left|y - f(\tilde{\theta}, x)\right|_2 \geq \log(\hat{p}^\alpha) - \frac{3q}{2}\log\left(\frac{2\pi}{n}\right) - \log\pi(\tilde{\theta}) + \frac{1}{2}\log|\mathcal{I}(\tilde{\theta})|$$

$$+ \log|\mathcal{I}_{\ell_D}(\hat{\theta})| - 2\ell_D(\hat{\theta}) - 2\log\pi(\hat{\theta}).$$

By Lemma 1, this entails that

$$\log\left|y - f(\hat{\theta}, x)\right|_2 \geq \log(\hat{p}^\alpha) - \frac{3q}{2}\log\left(\frac{2\pi}{n}\right) - \log\pi(\hat{\theta}) + \frac{1}{2}\log|\mathcal{I}(\hat{\theta})|$$

$$+ \log|\mathcal{I}_{\ell_D}(\hat{\theta})| - 2\ell_D(\hat{\theta}) - 2\log\pi(\hat{\theta})$$

$$= \log(\hat{p}^\alpha) - \frac{3q}{2}\log\left(\frac{2\pi}{n}\right) + \frac{1}{2}\log|\mathcal{I}(\hat{\theta})| + \log|\mathcal{I}_{\ell_D}(\hat{\theta})| - 2\ell_D(\hat{\theta}) - 3\log\pi(\hat{\theta}).$$

Notice that thanks to our choice of the loss, we have that both Fisher information matrices $\mathcal{I}(\tilde{\theta})$ and $\mathcal{I}_{\ell_D}(\hat{\theta})$ do not depend on $\hat{y}_{n+1}$. Similarly, the normalization term $\frac{3q}{2}\log\left(\frac{2\pi}{n}\right)$ and the prior values $\log\pi(\hat{\theta})$ and $-2\log\pi(\hat{\theta})$ do not depend on $\hat{y}_{n+1}$.[7] The Highest Density Region (HDR), then, is given by

---

[7]Neither directly through $y$ nor indirectly through $\tilde{\theta} = \arg\max_\theta \tilde{\ell}(\theta)$.

$$R(\hat{p}^\alpha) = \left\{ y \in \mathcal{Y} : \left| y - f(\hat{\theta}, x) \right|_2 \right.$$
$$\left. \geq \hat{p}^\alpha \exp\left[ \underbrace{-\frac{3q}{2} \log\left(\frac{2\pi}{n}\right) + \frac{1}{2} \log |\mathcal{I}(\hat{\theta})| + \log |\mathcal{I}_{\ell_D}(\hat{\theta})| - 2\ell_D(\hat{\theta})}_{=:\mathfrak{c}} - 3\log \pi(\hat{\theta}) \right] \right\}, \quad (27)$$

where we group in $\mathfrak{c}$ constants that do not depend on the prior. We can rewrite the expression for $\hat{p}^\alpha$ we derived before as

$$\hat{p}^\alpha = \sqrt[3]{\frac{9}{4}}(1-\alpha)^{\frac{2}{3}} \exp\left[ \frac{\mathfrak{c}}{3} + \log \pi(\hat{\theta}) \right].$$

Plugging this value in equation 27, we obtain $\hat{p}^\alpha_{\mathcal{L}}$,

$$\hat{p}^\alpha_{\mathcal{L}} = \hat{p}^\alpha \exp\left[ \mathfrak{c} - 3\log \pi(\hat{\theta}) \right] = \sqrt[3]{\frac{9}{4}}(1-\alpha)^{\frac{2}{3}} \exp\left[ \frac{\mathfrak{c}}{3} + \mathfrak{c} - 3\log \pi(\hat{\theta}) + \log \pi(\hat{\theta}) \right]$$
$$= \sqrt[3]{\frac{9}{4}}(1-\alpha)^{\frac{2}{3}} \exp\left[ \frac{4\mathfrak{c}}{3} - 2\log \pi(\hat{\theta}) \right],$$

which concludes the argument. The HDR, then, is

$$R(\hat{p}^\alpha) = \left\{ y \in \mathcal{Y} : \left| y - f(\hat{\theta}, x) \right|_2 \geq \sqrt[3]{\frac{9}{4}}(1-\alpha)^{\frac{2}{3}} \exp\left[ \frac{4\mathfrak{c}}{3} - 2\log \pi(\hat{\theta}) \right] \right\}$$

or, equivalently,

$$R(\hat{p}^\alpha) = \left\{ y \in \mathcal{Y} : \left| y - f(\hat{\theta}, x) \right|_2 \geq \sqrt[3]{\frac{9}{4}(1-\alpha)^2} \exp\left[ \frac{4\mathfrak{c}}{3} \right] \pi(\hat{\theta})^{-2} \right\}.$$

## B    Further Details on Experiments

We evaluate our method across synthetic and real-world benchmarks, comparing against strong probabilistic and deterministic baselines. Metrics cover both accuracy and uncertainty quality, including negative log-likelihood, Brier score, expected calibration error, and out-of-distribution (OOD) detection (via AUROC).[8] We ablate the contribution of each component of the approach and study sensitivity to hyperparameters and model scale. Robustness is assessed under distribution shift and label noise, and we report computational overhead (training/inference time and memory) relative to baselines. Overall, the method consistently matches or exceeds baseline accuracy while delivering better-calibrated uncertainties and competitive OOD performance at modest additional cost. Code to reproduce the experimental setup can be found at `https://github.com/rodemann/ssla`.

### B.1    Conjugate Prior Analysis

Conjugate Prior Analysis allows us to compare (A)SSLAs performance against the analytic PPD.

In the following, we briefly state each of the models and provide the experimental insights.

---

[8]For a modern treatment of OOD via semantic information, we refer the interested reader to Kaur et al. (2023).

### B.1.1 Normal-Normal Model

Consider a set of observations $X_1, \ldots, X_n \sim \mathcal{N}(\mu, \sigma^2_{\text{true}})$ where $\sigma^2_{\text{true}}$ is assumed to be known. The prior for the mean parameter $\mu$ is defined as $\mu \sim \mathcal{N}(\mu_0, \tau_0^2)$, where $\mu_0$ and $\tau_0^2$ are also known.

The posterior predictive distribution for a new observation $X_{new}$, given $X$, is then defined as

$$X_{new}|X \sim \mathcal{N}(\mu_n, \sigma_n^2 + \sigma_{true}^2) \tag{28}$$

where

$$\sigma_n^2 = \left( \frac{n}{\sigma_{true}^2} + \frac{1}{\tau_0^2} \right)^{-1}; \quad \mu_n = \left( \frac{\mu_0}{\tau_0^2} + \frac{\sum X_i}{\sigma_{true}^2} \right) \sigma_n^2 \tag{29}$$

In our experiment, we employ a true data-generating process that assumes parameter values of $\mu_{true} = 4.0$ and $\sigma_{true} = \sqrt{2.0}$, and the prior distribution is specified as $\mathcal{N}(\mu_0 = 4.0, \tau_0^2 = 1.0)$. Observations are sampled from $\mathcal{N}(\mu_{true}, \sigma_{true}^2)$, and the posterior predictive distribution is computed analytically according to Equation 28.

For SSLA and ASSLA, the log-likelihood and observed Fisher information matrix play a critical role in the approximations. Under our assumptions, the observations $X_i \sim \mathcal{N}(\mu_{true}, \sigma_{true}^2)$ for $i \in \{1, \ldots, n\}$ are independent and identically distributed. The log-likelihood is given by:

$$
\begin{aligned}
\ell(\mu) = \log \mathcal{L}(\mu) &= \log \prod_{i=1}^n f(X_i|\mu) = \sum_{i=1}^n \log f(X_i|\mu) \\
&= -\frac{n}{2} \log(2\pi\sigma^2) - \frac{1}{2\sigma^2} \sum_{i=1}^n (X_i - \mu)^2
\end{aligned} \tag{30}
$$

The observed Fisher information is defined as the negative second derivative of the log-likelihood:

$$\mathcal{J}(\mu) = -\frac{\partial \ell(\mu)}{\partial \mu, \mu} = -\frac{\partial}{\partial \mu} \frac{1}{\sigma^2} \sum_{i=1}^n (X_i - \mu) = \frac{n}{\sigma^2} \tag{31}$$

Figure 1 provides a comparison of the approximation quality of SSLA and ASSLA across various sample sizes ($n = 20, \ldots, n = 1.000.000$). For most cases ($n = 20, \ldots, n = 100.000$), both SSLA and ASSLA closely match the analytic posterior predictive distribution. This is further validated by the measured entropy between each approximation method and the analytic posterior predictive distribution, which remains close to zero across all cases. [9] These results confirm the ability of both methods to reconstruct the true distribution effectively.

### B.1.2 Poisson-Gamma Model

Let $X_1, \ldots, X_N \overset{iid}{\sim} Poi(\lambda)$, $\lambda \sim Gamma(\alpha, \beta)$. Then $\lambda|x \sim Gamma(\sum X_i + \alpha, n + \beta)$. The posterior predictive distribution for a new observation $X_{new}$ follows a Negative Binomial distribution:

$$X_{new}|X \sim NegBin\left( r = \sum X_i + \alpha, p = \frac{n+\beta}{n+\beta+1} \right)$$

with the definition of r being the number of successes, p being the success probability.

For the model setup, we assume a true data generating process of $X_i \sim Poi(\lambda_{true})$ for $i \in \{1, \ldots, n\}$ with a true rate of $\lambda_{true} = 3.0$.

The prior distribution is specified as a Gamma distribution $\lambda \sim Gamma(\alpha = 6.0, \beta = 2.0)$ where $\alpha$ and $\beta$ correspond to the prior's shape and rate parameters, respectively.

---

[9]A low entropy score indicates that the approximations do not introduce significant information loss relative to the analytic solution. An entropy of 0 tells us that we can use either technique interchangeably.

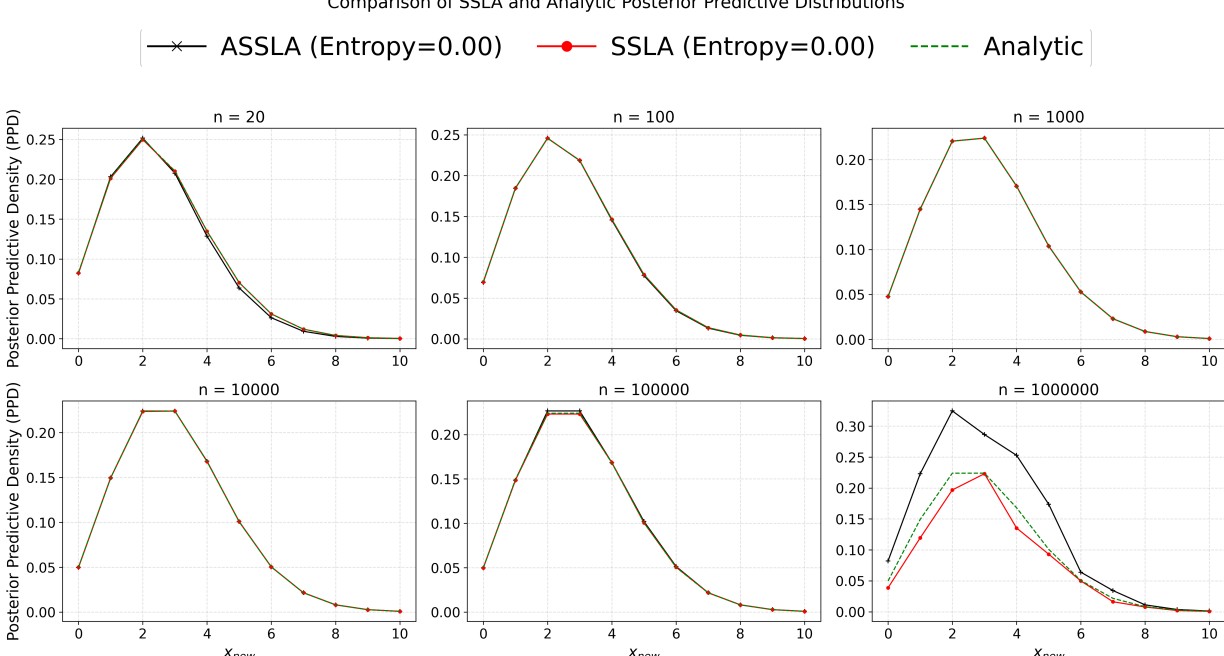

Figure 3: Illustrated are the analytic (green) and approximated (SSLA - red, ASSLA - black) PPDs for varying sample sizes $n$, ranging from $n = 20$ to $n = 1.000.000$. The results indicate close alignment for smaller and moderate data sizes, while deviations for larger sample sizes ($n = 1.000.000$) become noticeable for ASSLA.

We again derive the log-likelihood and the observed Fisher Information which are given by

$$\ell(\lambda) = \sum \left(X_i \log(\lambda) - \lambda\right) - \sum \log(X_i!) \tag{32}$$

and

$$\mathcal{J}(\lambda) = \frac{\sum X_i}{\lambda^2} \tag{33}$$

respectively.

The experiment evaluates the approximation quality of SSLA and ASSLA across different sample sizes ($n = 20, \ldots, n = 1.000.000$), see Figure 3. Observations are sampled from the true Poisson distribution, and the posterior predictive density for a new observation $X_{new}$ is computed using the three methods, (a) SSLA, (b) ASSLA, (c) Analytically.

## C   Heteroscedastic Regression: Further insights

### C.1   Architectural Design

We use a Multilayer Perceptron (MLP) (Bishop & Nasrabadi, 2006; Peng, 2017) as the base model due to its flexibility and effectiveness in modeling non-linear relationships.

The network architecture is formally defined as:

$$\hat{\mu}(x) = f_\theta(x) = W_n \cdot \text{act}\left(W_{n-1} \cdot \text{act}(\ldots (W_1 x + b_1)) + b_{n-1}\right) + b_n \tag{34}$$

for standard regression tasks, and as:

$$\begin{bmatrix} \hat{\mu}(x) \\ \log \hat{\sigma}^2(x) \end{bmatrix} = f_\theta(x) = W_n \cdot \text{act}\left(W_{n-1} \cdot \text{act}(\dots (W_1 x + b_1)) + b_{n-1}\right) + b_n \qquad (35)$$

for modeling heteroscedastic uncertainty, where the output consists of both the predicted mean and the logarithm of the variance.

We employ two hidden layers with 50 neurons each and ReLU activation for modeling non-linear relationships.

## C.2 Hyperparameter Optimization

To optimize the architectural design of the network, we use `Optuna` (Akiba et al., 2019). Optuna is a flexible hyperparameter optimization framework that dynamically constructs the search space and leverages Bayesian Optimization (BO) (Frazier, 2018), specifically using the Tree-structured Parzen Estimator (TPE) (Bergstra et al., 2011) approach for single-objective optimization.

The following parameters were optimized based on validation loss:

- Learning Rate: $10^{-5}, \dots, 10^{-1}$
- Batch Size: $32, \dots, 256$

The search was conducted for 100 trials, after which the best hyperparameters were selected.

**HMC - Hyperparameter Tuning**  For HMC we tune the following hyperparameters:

- Step Size
  - Description: The size of steps taken during the simulation of the Hamiltonian dynamics. It relates to the exploration of the sampler (Cobb, 2023a)
  - Search Space: $10^{-5}, \dots, 10^{-3}$
- Number of Samples
  - Description: This reflects the total number of samples generated from the posterior distribution
  - Search Space: $500, \dots, 2000$
- Number of Steps per Sample
  - Description: The number of leapfrog steps taken in the Leapfrog integration technique. For further information on leapfrog integration, used in HMC we refer the reader to Pourzanjani & Petzold (2019).
  - Search Space: $10, \dots, 50$
- Tau In
  - Description: This parameter reflects the precision the prior (Cobb, 2023a)
  - Search Space: $0.1, \dots, 10$
- Tau Out
  - Description: This parameter reflects likelihood output precision (Cobb, 2023a)
  - Search Space: $10, \dots, 500$
- Mass
  - Description: In HMC the mass is typically encoded as a mass matrix (Betancourt, 2018) and directly impacts the step size and trajectory of the leapfrog integrator. In hamiltorch, a diagonal matrix with scaling factor (i.e. the `mass` parameter) is used (Cobb, 2023a)
  - Search Space: $0.1, \dots, 10$

**BNN-MFVI**  For the BNN with Mean-Field Variational Inference approximation the following hyperparameters are tuned

- Burnin Epochs

    - Description: Represents the number of epochs to train before switching to NLL loss
    - Search Space: $50, \ldots, 200$

- Number of MC Samples During Train

    - Description: The number of MC Samples to draw during training when computing the negative ELBO loss
    - Search Space: $5, \ldots, 50$

- Number of MC Samples During Test

    - Description: The number of MC Samples to draw during test and prediction
    - Search Space: $10, \ldots, 100$

- Output Noise Scale

    - Description: The scale of the predicted sigmas
    - Search Space: $0.5, \ldots, 2$

**BNN-VI**  The BNN-VI model is proposed in (Depeweg et al., 2018) and requires next to the hyperparameters for the MLP architecture and the hyperparameters of MFVI additionally:

- Alpha

    - Description: This parameter is used to minimize the ($\alpha$-) divergence (see e.g. Minka, 2005) between the variational and the analytic posterior (Depeweg et al., 2018).
    - Search Space: $0, \ldots, 1$

**LA - Hyperparameter Tuning**  The Laplace class provided by Daxberger et al. (2021a) enables automatic post-hoc tuning of the prior precision using the marginal likelihood method (Immer et al., 2021a), as described by Daxberger et al. (2021a, Chapter 3). We follow their recommendation (Daxberger et al., 2021a, Chapter 4.1) in applying post-hoc tuning rather than online training.

### C.3  Loss Functions

Depending on the regression task and the UQ method, we employ different loss functions:

- Mean Squared Error (MSE) Loss: Used for deterministic regression models

$$\mathcal{L}_{MSE} = \frac{1}{n} \sum_{i=1}^{n} (y_i - \mu(x_i))^2$$

- Negative Log-Likelihood (NLL): Used for models predicting both mean and variance

$$\mathcal{L}_{NLL} = \frac{1}{2} \sum_{i=1}^{n} \left( \log\big(\sigma(x_i)^2\big) + \frac{(y_i - \mu(x_i))^2}{\sigma(x_i)^2} \right)$$

## C.4 Training Configuration

- Optimizer: Adam with a learning rate according to the found learning rate hyperparameter

- Training Epochs: We used 500 epochs of training with early stopping after 20 epochs of no improvement in the validation loss

- Batch Size: The batch size is adapted according to the found hyperparameter

- Monte Carlo Samples: For (A)SSLA we used 20 samples to reconstruct the ppd of each observation, varying the response $y$ by small margins around the true predicted $\hat{y}$

- We assume a standard normal distributed prior where applicable

# D  Additional Experiments

Tables 5 and 6 have some additional results on the Lightning UQ benchmarks (Lehmann et al., 2024). They comprise the Sine RBF (Radial basis function) setup and compare the homoscedastic case to the heteroscedastic RBF regression case. As expected from the theory, SSLA and ASSLA coincide with classical Laplace-style behavior when refitting effects are negligible, thereby validating the asymptotic approximations underlying ASSLA.

| Method | RMSE ↓ | NLL ↓ | Cov90 ↑ | Cov95 ↑ |
|---|---|---|---|---|
| Deterministic (plugin) | $0.310 \pm 0.042$ | $0.316 \pm 0.234$ | $82.0 \pm 9.9$ | $91.0 \pm 9.9$ |
| Laplace/Exact | $0.310 \pm 0.042$ | $0.307 \pm 0.219$ | $85.5 \pm 9.2$ | $92.0 \pm 9.9$ |
| ASSLA | $0.310 \pm 0.042$ | $0.316 \pm 0.235$ | $82.0 \pm 9.9$ | $91.0 \pm 9.9$ |
| SSLA | $0.310 \pm 0.042$ | $0.307 \pm 0.219$ | $85.5 \pm 9.2$ | $92.0 \pm 9.9$ |

Table 5: Lightning UQ Box benchmark: Sine (homoscedastic) — RBF regression.

| Method | RMSE ↓ | NLL ↓ | Cov90 ↑ | Cov95 ↑ |
|---|---|---|---|---|
| Deterministic (plugin) | $0.346 \pm 0.034$ | $0.420 \pm 0.182$ | $87.0 \pm 4.2$ | $89.5 \pm 4.9$ |
| Laplace/Exact | $0.346 \pm 0.034$ | $0.413 \pm 0.167$ | $87.5 \pm 4.9$ | $90.0 \pm 4.2$ |
| ASSLA | $0.346 \pm 0.034$ | $0.420 \pm 0.182$ | $87.0 \pm 4.2$ | $89.5 \pm 4.9$ |
| SSLA | $0.346 \pm 0.034$ | $0.413 \pm 0.167$ | $87.5 \pm 4.9$ | $90.0 \pm 4.2$ |

Table 6: Lightning UQ Box benchmark: Sine (heteroscedastic) — RBF regression.

# E  Illustration of prior modularity:

We illustrate the prior modularity on a simple 1D linear regression toy problem with $n = 50$ and Gaussian noise $\sigma = 0.5$. We evaluate SSLA/ASSLA at $x_{\text{test}} = 1.5$. For each prior $w \sim \mathcal{N}(0, \tau^2)$ we compute $\hat{\theta}$ and the self-prediction $\hat{y}_\star = f_{\hat{\theta}}(x_{\text{test}})$, then evaluate the log-PPD approximation at (A) $y = \hat{y}_\star$ and (B) $y = \hat{y}_\star + 1.0$. We include a near-noninformative prior $\tau^2 = 10^6$.

| Prior | SSLA | ASSLA |
|---|---|---|
| $\mathcal{N}(0, 0.01)$ | $-0.0259$ | $-0.0259$ |
| $\mathcal{N}(0, 0.1)$ | $-0.0259$ | $-0.0259$ |
| $\mathcal{N}(0, 1)$ | $-0.0259$ | $-0.0259$ |
| $\mathcal{N}(0, 10)$ | $-0.0259$ | $-0.0259$ |
| $\mathcal{N}(0, 10^6)$ | $-0.0259$ | $-0.0259$ |

Table 7: At the self-prediction $y = \hat{y}_\star$, SSLA and ASSLA coincide (up to constants), since the pseudo-observation induces no predictive "surprise" at $\hat{y}_\star$.

| Prior | SSLA | ASSLA | SSLA–ASSLA | SSLA–SSLA$_{\mathrm{noninfo}}$ |
|---|---|---|---|---|
| $\mathcal{N}(0, 0.01)$ | $-1.9612$ | $-2.0259$ | $0.0647$ | $-0.0363$ |
| $\mathcal{N}(0, 0.1)$ | $-1.9303$ | $-2.0259$ | $0.0956$ | $-0.0054$ |
| $\mathcal{N}(0, 1)$ | $-1.9255$ | $-2.0259$ | $0.1004$ | $-0.0006$ |
| $\mathcal{N}(0, 10)$ | $-1.9250$ | $-2.0259$ | $0.1009$ | $-0.0001$ |
| $\mathcal{N}(0, 10^6)$ | $-1.9250$ | $-2.0259$ | $0.1009$ | $0.0000$ |

Table 8: Off the self-prediction $y = \hat{y}_\star + 1.0$, tightening the prior (smaller variance) changes SSLA, while ASSLA is (by construction) much less sensitive at the approximation order where the prior increment is dropped.

