# OpenReview forum: "Self-Supervised Laplace Approximation for Bayesian Uncertainty Quantification"
_TMLR — Accepted by TMLR_

### Review · Reviewer_z9r6 · 2025-12-09

**Summary Of Contributions:**

This work proposes the Laplace Approximation method for effectively estimating the posterior predictive distribution for uncertainty quantification in the Bayesian deep learning settings. The main proposal is achieved via a self-supervised learning framework. Multiple evaluations are performed to validate the methods.

**Additional Comments:**

N/A

**Audience:**

Yes

**Audience Explanation:**

1. Laplace approximation in Bayesian deep learning is a highly relevant and active research topic, especially for uncertainty quantification.

2. The idea presented is novel, especially for regression tasks.

**Claims And Evidence:**

Yes

**Claims Explanation:**

1. The motivation and background of the work are clearly stated.

2. The theoretical proofs are provided in the Appendices.

3. Multiple case evaluations are performed to validate the method, the experimental configurations are well documented.

**Requested Changes:**

1. In terms of presentation, I would suggest having the pseudo-code for the algorithms in the main body to improve the clarity of the methods and readability. A high-level conceptual figure of the methods might also be for consideration. Is the regression task the focus of the proposed methods? If yes, it is better to be clear in the main body, including the abstract. Otherwise, a case study on classification may be needed.

2. The method requires the log-likelihood of the predicted instance. How would the method guarantee that the log-likelihood is reliable? E.g., a reliable model (in terms of training metrics) can still make unreliable predictions (for example, in the case of serve data drift), which could result in unreliable log-likelihood?

3. Although multiple experiments are conducted, the tasks are relatively simple and a bit outdated. There are some modern UQ benchmarks worth considering [1]. The comparison baseline methods can also be rich.

[1] Lightning UQ Box: A Comprehensive Framework for Uncertainty Quantification in Deep Learning,

4. It is claimed that *the method is modular with respect to the prior specification*. It will be helpful to include some validations in this aspect, in comparison with some mentioned literature, e.g., Credal Bayesian Deep Learning (Caprioel al. 2024).

---

> ### Author Response · Authors · 2026-01-09
>
> We thank the reviewer for the encouraging evaluation and for the constructive suggestions on strengthening the presentation and experimental relevance. We are glad the manuscript was perceived as addressing an active and important topic and that the central idea was viewed as novel, sound and well documented. What is more, we are very grateful for the concrete and actionable advice on how to improve our manuscript.
>
> ### Summary of the suggested improvements in the revised version:
>
> We implement the reviewer’s recommendations as follows.
>
> - **Add pseudo-code and code:** We agree this is important. The current presentation of our method is equation-dense, and including algorithms will make the method easier to understand and implement correctly. We will also include key computational steps (refit vs no-refit, curvature) in the pseudo-code. Beside, we go even further and not only share pseudo-code but also the entire codebase of our method on an anonymized repository: https://anonymous.4open.science/r/ssla
>
> - **Clarify scope (regression focus) and classification discussion:** We agree that the scope should be more explicit in the abstract and introduction. In the revision, we emphasize  that our scope is regression. We will make that clear and add a dedicated discussion on how the approach extends to classification (and what changes are required). If space allows, we will include a small classification case study; otherwise, we will provide a precise limitation statement and concrete directions for extension.
>
> - **Reliability of log-likelihood under drift / misspecification:** We will expand the limitations discussion to clarify that SSLA/ASSLA (just like any other approximate Bayesian inference method) depend on the quality of the likelihood model and may inherit misspecification errors. We will also add practical guidance (e.g., combining with OOD detection / robust likelihood modeling).
>
> - **Lightning UQ benchmarks:** We appreciate the suggestion and will expand the evaluation where feasible. In the revision, we mention more prominently that we are already using the suggested Lightning UQ box (see Section 5.1 in the original submission). In addition to extending the experiments, we will clarify the trade-offs in runtime and complexity, allowing readers to interpret performance in the context of computational budget.
>
> - **Explain and Illustrate Prior modularity:** We agree that both a more in-depth discussion and a targeted prior-sensitivity experiment will make this claim more concrete. We will add an explicit experiment varying prior strength/structure and report the effect on predictive uncertainty and calibration.
>
> $~$
>
> We believe these revisions will significantly improve accessibility and strengthen the empirical story, while the underlying method and supporting theory remains unchanged. Given the reviewer’s positive assessment, we hope the reviewer will agree to accept the paper after we upload the improved version

---

> ### Author Response · Authors · 2026-01-09
>
> ### Please find more detailed answers to the reviewer’s questions and requested changes below:
>
> $~$
>
> > In terms of presentation, I would suggest having the pseudo-code for the algorithms in the main body to improve the clarity of the methods and readability. A high-level conceptual figure of the methods might also be for consideration. I
>
> Many thanks for suggesting this. We share **both code and pseudo-code of SSLA and ASSLA**. We completely agree that adding pseudo-code can increase the accessibility of the paper. Besides adding the following pseudo-code in an algorithm environment to the manuscript, we also share the entire implementation of our methods via the anonymized repository: https://anonymous.4open.science/r/ssla
>
> > Is the regression task the focus of the proposed methods? If yes, it is better to be clear in the main body, including the abstract. Otherwise, a case study on classification may be needed.
>
> We thank the reviewer for this question. Yes, regression is indeed the focus of SSLA and ASSLA. We fully agree that this deserves a more prominent statement clarifying that we explicitely focus on regression as opposed to classification.
>
> (The reason is the research gap explained on page 3 and 4: “. [...] Daxberger et al. (2021a) summarize and implement a few alternatives ranging from probit approximation (Spiegelhalter & Lauritzen, 1990) for binary classification to extended probit (Gibbs, 1998) or approximating the softmax-Gaussian integral via a Dirichlet distribution (Hobbhahn et al., 2022) for general classification. In the regression case, however, there is no established method on how to circumvent expensive MC-sampling apart from linearization of the network (Daxberger et al., 2021b; Immer et al., 2021b)”)
>
> In the revision, we emphasize this focus on regression setups more prominently in the abstract as well as additionally in the main paper on pages 2, 7, 8 and 9. We agree entirely with the reviewer that this is crucial  to clearly limit the scope to regression and provide a more precise discussion of what would be required for classification, as detailed below.
> For the sake of completeness, what happens in the classification case is the following: The predictive is $p(y\mid x,D)=\int Softmax_y(g_\theta(x))\,p(\theta\mid D)\,d\theta,$ and Laplace approximation are often paired with additional approximations (e.g., probit/softmax integral approximations).
> Crucially, SSLA/ASSLA can still be applied to approximate $\log p(\hat y\mid x,D)$ for a pseudo-label $\hat y$ (e.g., argmax class or soft label), but the interpretation and calibration questions differ because $\hat y$ is discrete and the likelihood differs.

---

> > ### Author Response · Authors · 2026-01-09
> >
> > > The method requires the log-likelihood of the predicted instance. How would the method guarantee that the log-likelihood is reliable? E.g., a reliable model (in terms of training metrics) can still make unreliable predictions (for example, in the case of serve data drift), which could result in unreliable log-likelihood?
> >
> >
> > We thank the reviewer for raising this point. Checking model specifications/assumptions is crucial for OOD applications (like dataset shift, covariate shift, or more general forms of model misspecification.) Our response is structured in  three parts: (i) what the method does not guarantee, (ii) what the method does guarantee conditional on the likelihood model, and (iii) how SSLA/ASSLA should be interpreted and used in practice under potential unreliability.
> >
> > (i) Obviously, SSLA and ASSLA do not guarantee that the model specification (i.e., the chosen loss/likelihood) and thus the predictive log-likelihood is “reliable” in an absolute sense. This is unavoidable: any Bayesian or likelihood-based uncertainty quantification method fundamentally inherits the assumptions of the likelihood model. If the likelihood is misspecified, or if the test input lies far outside the training distribution (e.g., under severe data drift), then the numerical value of $\log p(\hat y_{n+1}\mid x_{n+1},\hat\theta)$
> > may indeed be misleading. This limitation is not specific to SSLA/ASSLA, but shared by any type of (approximate) Bayesian inference method like Laplace approximations, variational inference, MCMC, deep ensembles etc.
> >
> > (ii) What SSLA/ASSLA do guarantee is the following, conditional on the assumed likelihood being a meaningful local model around (\hat\theta):
> >
> > * The method does not rely on the absolute value of the log-likelihood alone. Instead, predictive uncertainty is quantified through changes induced by the pseudo-observation, namely:
> >   - the change in the objective (likelihood plus prior), and
> >   - the change in local curvature (via the Fisher or Hessian log-determinant).
> > * In particular, the score is driven by how sensitive the fitted model and its local geometry are to the inclusion of the self-predicted instance. Intuitively, if a prediction is fragile—i.e., small perturbations induced by the pseudo-observation lead to large shifts in curvature or objective—SSLA/ASSLA assign higher predictive uncertainty.
> > * Note that his mechanism can amplify warning signals even when the raw likelihood value itself is overconfident. Under drift or misspecification, pseudo-observations often induce atypical curvature behavior (e.g., poorly conditioned Fisher matrices or large log-det changes), which is reflected directly in the SSLA/ASSLA score. Thus, SSLA/ASSLA are not, if you allow, “blind” to unreliability: they operationalize a local robustness test of the model around the test point, rather than trusting the likelihood alone.
> >
> > (iii) We agree with the reviewer that, under severe dataset shift, any likelihood-based score must be treated with caution. In such regimes, SSLA/ASSLA should be interpreted as conditional uncertainty estimates: they are informative insofar as the likelihood provides a reasonable local description of the data-generating process near (x_{n+1}).
> > In the revised manuscript, we will therefore:
> > * Explicitly how SSLA/ASSLA behave under likelihood misspecification in the limitations section.
> > * Emphasize that SSLA/ASSLA are complementary to (not replacements for) drift detection or OOD detection mechanisms.
> >
> > In summary, SSLA/ASSLA do not guarantee likelihood reliability under arbitrary misspecification (no Bayesian method can). Our contribution lies in translating the model’s local self-consistency to a prediction into a principled, computationally efficient uncertainty quantification method, which can still be informative even when absolute likelihood values are imperfect.

---

> ### Author Response · Authors · 2026-01-09
>
> > Although multiple experiments are conducted, the tasks are relatively simple and a bit outdated. There are some modern UQ benchmarks worth considering [1]. The comparison baseline methods can also be rich.
> [1] Lightning UQ Box: A Comprehensive Framework for Uncertainty Quantification in Deep Learning,
>
>
> We thank the reviewer for this remark. We completely agree that Lightning UQ Box offers modern and state-of-the-art UQ benchmarks. In fact, this is why we already use it, see Section 5.1 in the original submission. But we fully agree we could add even more experiments along this line. While some of them are still running at the time of writing, please find some preliminary results below. They are from the Sine RBF (Radial basis function) setup in LIghtning UQ and compare the homoscedastic case to the heteroscedastic RBF regression case. As expected from the theory, SSLA and ASSLA coincide with classical Laplace-style behavior when refitting effects are negligible, thereby validating the asymptotic approximations underlying ASSLA.
>
> ### Additional Lighting UQ Box benchmark: Sine (homoscedastic) RBF regression
> | Method | RMSE ↓ | NLL ↓ | Cov90 ↑ | Cov95 ↑ |
> |:--|--:|--:|--:|--:|
> | Deterministic (plugin) | 0.310 ± 0.042 | 0.316 ± 0.234 | 82.0 ± 9.9 | 91.0 ± 9.9 |
> | Laplace/Exact | 0.310 ± 0.042 | 0.307 ± 0.219 | 85.5 ± 9.2 | 92.0 ± 9.9 |
> | ASSLA | 0.310 ± 0.042 | 0.316 ± 0.235 | 82.0 ± 9.9 | 91.0 ± 9.9 |
> | SSLA | 0.310 ± 0.042 | 0.307 ± 0.219 | 85.5 ± 9.2 | 92.0 ± 9.9 |
>
>
> ### Additional Lightning UQ Box benchmark: Sine (homoscedastic) RBF regression
>
> | Method | RMSE ↓ | NLL ↓ | Cov90 ↑ | Cov95 ↑ |
> |:--|--:|--:|--:|--:|
> | Deterministic (plugin) | 0.346 ± 0.034 | 0.420 ± 0.182 | 87.0 ± 4.2 | 89.5 ± 4.9 |
> | Laplace/Exact | 0.346 ± 0.034 | 0.413 ± 0.167 | 87.5 ± 4.9 | 90.0 ± 4.2 |
> | ASSLA | 0.346 ± 0.034 | 0.420 ± 0.182 | 87.0 ± 4.2 | 89.5 ± 4.9 |
> | SSLA | 0.346 ± 0.034 | 0.413 ± 0.167 | 87.5 ± 4.9 | 90.0 ± 4.2 |
>
>
> In the revision, we include those results, mention more prominently that we are already using the suggested Lightning UQ box and we will include further additional benchmarks from Lightning UQ.

---

> > ### Author Response · Authors · 2026-01-09
> >
> > > It is claimed that the method is modular with respect to the prior specification. It will be helpful to include some validations in this aspect, in comparison with some mentioned literature, e.g., Credal Bayesian Deep Learning (Caprioel al. 2024).
> >
> > We thank the Reviewer for their advice. We agree that we could have done a better job at explaining what our method being modular with respect to the prior specification actually means. We now comment on that, and our argument will be added to the updated version of our paper.
> >
> > In Credal Bayesian Deep Learning (CBDL), the scholar is free to choose a finite collection of, say, $K$ priors (to account for possible prior misspecification), and similarly, a finite collection of, say, $J$ likelihoods (to account for possible likelihood misspecification and distribution drift). They are then combinatorially combined to derive $K \times J$ VI approximations of the (parameter) posteriors. In turn, the latter distributions are used to derive $K \times J$ VI approximations of the posterior predictive distributions. The convex hull of those posterior predictives is used to quantify predictive aleatoric (irreducible) and epistemic (reducible) uncertainties.
> >
> > The Laplace approximation method that we derive in this paper relates to the CBDL framework as follows. We focus on prior misspecification and treat the likelihood as fixed, i.e., $J=1$ in the CBDL notation. To do that, we can specify a finite collection of, say, $K$ priors as in CBDL. Together with the likelihood, each such prior is used to derive a posterior predictive without the double VI approximation needed in CBDL. We then end up with $K$ posterior predictives, whose convex hull is used for uncertainty quantification and downstream tasks.
> > As we can see, our method corresponds to CBDL for which $J=1$, with enormous savings with regard to  computational complexity.
> >
> > To illustrate this further, we include a toy prior-modularity example (1D linear regression) in the revised version. The setup is a very simple synthetic 1D regression with $n=50$ and Gaussian noise $\sigma=0.5$. We evaluate SSLA/ASSLA at a fixed test input $x_{\text{test}}=1.5$. For each prior, we first compute the $\hat\theta$ and the self-prediction $\hat y_\star=f_{\hat\theta}(x_{\text{test}})$. We then evaluate the log-PPD approximation at (i) $y=\hat y_\star$ and (ii) a nearby value $y=\hat y_\star+1.0$. Priors are $w\sim\mathcal N(0,\tau^2)$, including a near-noninformative prior $\tau^2=10^6$.
> >
> > As can be seen in the table below, in this toy setting, tightening the prior (smaller $\tau^2$) has a visible effect on SSLA away from $\hat y_\star$, while ASSLA is (by construction) essentially insensitive to the prior since the prior increment is dropped in the ASSLA simplification, see page 7 in the original submission.
> >
> > $~$
> >
> > ### At the self-prediction $y=\hat y_\star$
> >
> > | Prior      |    SSLA |   ASSLA |
> > |:-----------|--------:|--------:|
> > | N(0,0.01)  | -0.0259 | -0.0259 |
> > | N(0,0.1)   | -0.0259 | -0.0259 |
> > | N(0,1)     | -0.0259 | -0.0259 |
> > | N(0,10)    | -0.0259 | -0.0259 |
> > | N(0,1e+06) | -0.0259 | -0.0259 |
> >
> > ### Off the self-prediction $y=\hat y_\star + 1.0$
> >
> > | Prior      |    SSLA |   ASSLA |   SSLA-ASSLA |   SSLA-SSLA_noninfo |
> > |:-----------|--------:|--------:|-------------:|--------------------:|
> > | N(0,0.01)  | -1.9612 | -2.0259 |       0.0647 |             -0.0363 |
> > | N(0,0.1)   | -1.9303 | -2.0259 |       0.0956 |             -0.0054 |
> > | N(0,1)     | -1.9255 | -2.0259 |       0.1004 |             -0.0006 |
> > | N(0,10)    | -1.9250 | -2.0259 |       0.1009 |             -0.0001 |
> > | N(0,1e+06) | -1.9250 | -2.0259 |       0.1009 |              0.0000 |

---

> > > ### Author Response · Authors · 2026-01-09
> > >
> > > We also point out how our method can be used to derive a regression version of Interval Deep Evidential Classification in https://arxiv.org/abs/2512.05526. To see this, note that we can specify a single prior and likelihood (so $K=J=1$), apply our Laplace approximation method, and derive the approximated posterior predictive distribution. Then, we denoting by $\ell$ the pdf of the latter, iIt induces an interval of measures (a tool from the Imprecise Probability theory literature) $[\ell, (1+d) \ell ]$, $d \geq 0$. Such an interval is one-to-one with a convex (and closed) set of posterior predictive probabilities (Walley, 1991: “Statistical Reasoning with Imprecise Probabilities”, Section 4.6.4), which can in turn be used for uncertainty quantification and downstream tasks.
> > >
> > > A corollary to (Walley, 1991, Section 4.6.4) tells  that we can always find a collection of $K$ priors and of $J$ likelihoods (allowing also either $K$ or $J$, but not both, to be equal to $1$) such that the convex hull of the $K \times J$ resulting (non-approximated) posterior predictives corresponds to the set of probabilities in a one-to-one relationship with $[\ell, (1+d) \ell ]$.
> > >
> > > Notice furthermore that a procedure to find an optimal $d^\star \geq 0$ exists, where optimal has to be understood as minimizing the coverage discrepancy between the precise posterior predictive distribution and the interval of measures. Such a discrepancy is inevitable because we go from the precise (only one distribution) to the imprecise (a set of distributions) case.
> > >
> > > $~$
> > >
> > > **We thank the reviewer once more for helping us improve our manuscript by their detailed and constructive remarks and questions. We believe it really did. Please kindly let us know if you have any further questions or concerns. We are happy to provide further clarifications.**

---

### Review · Reviewer_yXWb · 2025-12-23

**Summary Of Contributions:**

The paper introduces two methods, Supervised Laplace Approximation (SSLA) and Approximate Self-Supervised Laplace Approximation (ASSLA). Both aim to perform approximate Bayesian inference while avoiding the direct computation of the posterior parameter distribution, instead approximating the posterior predictive distribution directly.
In the case of SSLA, predictive uncertainty is quantified by refitting the model on an augmented dataset that includes the model’s own predictions. However, this solution has additional overhead because of the refitting phase. Therefore, the authors introduce a second method, ASSLA, that avoids the refitting phase by using asymptotic expansions and local linearization to express the posterior predictive distribution.

The authors compare their methods with three other approximate Bayesian inference techniques using both a synthetic dataset and datasets from the UCI Machine Learning Repository.

The paper also discusses some scalability and stability issues. The authors report that ASSLA suffers from numerical instabilities when the sample size exceeds n=1.000.000. SSLA, instead, is unstable on larger datasets and in the presence of prior–data conflicts.

**Audience:**

Yes

**Audience Explanation:**

I believe the contribution is sound and that it could be beneficial for researchers working in the field.

**Broader Impact Concerns:**

I do not have any ethical concerns about this work

**Claims And Evidence:**

Yes

**Claims Explanation:**

The claims made in the submission are convincing. The authors clearly explain the proposed methods and the results obtained. They also explain the limitations of their methods, briefly discussing them. I believe the paper would be stronger with a more extensive and in-depth discussion of these limitations.

**Requested Changes:**

- The plots can be improved: (1) the font of the ticks, axes, titles, and legends is too small, the authors should try increasing it. (2) In general, there is a lack of consistency across the plots: Figures 1 and 3 have the legend inside each plot, even though the legend is always the same (and it is also hard to read because of the small font). Figure 3, instead, has the legend on top of the plots. The same issue applies to the plot titles: Figures 1 and 3 have a title, while Figure 2 does not.
- On page 7, the authors refer to Figure 3. However, this figure is in the appendix. Figure 1, instead, is not referred to in the main paper, but it is referred in the appendix. I guess these two references need to be swapped.
- On page 7, the authors say that SSLA and ASSLA are able to recover the analytical PPD almost perfectly for sample sizes between n=20 and n=100,000. Then, in the same line, they state that numerical instabilities emerge for n >= 1,000,000. What happens in the range between 100,000 and 1,000,000?
- The authors should elaborate more on the numerical instability issues.
- The authors state that ASSLA balances computational tractability with competitive probabilistic calibration. At the same time, in Section 5.1, the experiments show that it underestimates uncertainty over wider intervals, leading to a risk-seeking bias. Would it be possible to elaborate more on this?
- In Section C.2 (page 26 in the appendix), the authors report the tuned hyperparameters. The values of these parameters are written as “1e-5”, I am not sure whether the LaTeX was rendered correctly here.
- In Section C.2 of the appendix, there are some question marks inside parentheses (?) at the end of some sentences.

---

> ### Author Response · Authors · 2026-01-09
>
> We thank the reviewer for the supportive assessment and the thoughtful, concrete suggestions. We are glad that the manuscript was perceived as technically sound and useful for researchers interested in scalable approximate Bayesian inference (ABI) and uncertainty quantification. We also appreciate the reviewer’s careful attention to presentation quality and to the numerical stability discussion. Furthermore, we fully agree that the paper can be improved (and we are optimistic it has improved) by discussing these limitations in greater detail, as suggested by the reviewer.
>
> ### Summary of the suggested improvements in the revised version:
>
> - **Figures and readability:** We standardize the plotting style (larger fonts, consistent titles, consistent legend placement). We agree these changes will significantly improve clarity, especially for readers skimming results and comparing methods. We also fix the cross-referencing issues between Figure 1 and 3. Thanks for spotting!
>
> - **Intermediate regime $n\in[10^5,10^6]$:** We ran additional experiments with $n$ between $10^5$ and $10^6$: They show that the approximation is still stable but not perfectly matching the analytical PPD.
>
> - **Instability discussion:** We now will elaborate on *why* SSLA and ASSLA can become unstable in certain regimes (curvature conditioning, log-det sensitivity, numerical precision), and we will provide concrete mitigation strategies (damping in Cholesky, structured curvature approximations where applicable). We will also clarify which mitigations are used in our experiments to improve reproducibility by linking an (anomyzed) repository containing code to reproduce all experiments: https://anonymous.4open.science/r/ssla
>
> - **Calibration vs underestimation at wide intervals:** We fully agree that the text should more carefully reconcile these points and explain ASSLA’s coverage in the experiments in greater detail. Werevise the wording to be precise about the regimes where ASSLA underestimates uncertainty (and SSLA should be preferred).
>
> These requested changes are largely concerned with presentation and empirical clarification, and we believe they can be addressed cleanly in a revision without altering the core contribution. We hope the reviewers will agree to accept the paper after we upload the improved version.

---

> ### Author Response · Authors · 2026-01-09
>
> ### Please find more detailed answers to the reviewer’s questions and requested changes below:
>
> > The plots can be improved: (1) the font of the ticks, axes, titles, and legends is too small, the authors should try increasing it. (2) In general, there is a lack of consistency across the plots: Figures 1 and 3 have the legend inside each plot, even though the legend is always the same (and it is also hard to read because of the small font). Figure 3, instead, has the legend on top of the plots. The same issue applies to the plot titles: Figures 1 and 3 have a title, while Figure 2 does not.
>
> We fully agree that the plots can benefit from some polishing! We recreated all plots and tried to harmonize legend style and whether they have titles or not. We also increased the readability of the ticks, axes, titles and legends in Figure 1 as requested.
>
> > On page 7, the authors refer to Figure 3. However, this figure is in the appendix. Figure 1, instead, is not referred to in the main paper, but it is referred in the appendix. I guess these two references need to be swapped.
>
> Good catch! We correct the cross-references and ensure the narrative in the main text matches the figure placement. We agree that the current mismatch creates unnecessary friction for the reader and are thankful to the reviewer for spotting it!
>
> > On page 7, the authors say that SSLA and ASSLA are able to recover the analytical PPD almost perfectly for sample sizes between n=20 and n=100,000. Then, in the same line, they state that numerical instabilities emerge for n >= 1,000,000. What happens in the range between 100,000 and 1,000,000?
>
> For $n$ larger 100000 and smaller $10^6$, the approximation is still stable but (as indicated by the case of $n=10^6$ in Figure 1) not perfectly matching the analytical PPD. To clarify what happens between $n=10^5$ and $n=10^6$, we ran additional experiments (see our anonymized repository: https://anonymous.4open.science/r/ssla) in the Normal–Normal conjugate setting used in the paper. For each $n$, we generate six independent datasets and evaluate the predictive density on a 51-point grid on $[0,10]$. The goal is to isolate the numerical effect discussed in the paper: computing the predictive increment in ASSLA by subtracting two large $\mathcal{O}(n)$ log-likelihood aggregates in float32, versus a numerically stable computation of the same quantity (as effectively used by SSLA).
>
> **Metrics**
> - **KL**: discrete KL divergence between the stable reference curve and the ASSLA float32 curve.
> - **Max |log error|**: maximum pointwise absolute difference in log density on the grid.
> - **Frac(delta == 0)**: fraction of grid points where the float32 subtraction collapses the increment to exactly zero.
> - **Frac(nonfinite)**: fraction of grid points producing NaN or Inf.
>
> Each entry in the table has mean ± standard deviation over 6 runs.
>
> | n | runs | KL (mean ± sd) | Max \|log error\| (mean ± sd) | Frac(delta == 0) (mean ± sd) | Frac(nonfinite) (mean ± sd) |
> |---:|---:|---:|---:|---:|---:|
> | 100,000 | 6 | 9.41e-06 ± 1.3e-06 | 7.57e-03 ± 3.1e-04 | 0.00e+00 ± 0.0e+00 | 0.00e+00 ± 0.0e+00 |
> | 200,000 | 6 | 5.02e-05 ± 2.3e-06 | 1.55e-02 ± 2.7e-05 | 0.00e+00 ± 0.0e+00 | 0.00e+00 ± 0.0e+00 |
> | 300,000 | 6 | 1.41e-04 ± 3.3e-06 | 3.05e-02 ± 5.1e-04 | 0.00e+00 ± 0.0e+00 | 0.00e+00 ± 0.0e+00 |
> | 500,000 | 6 | 1.40e-04 ± 3.3e-06 | 3.06e-02 ± 3.1e-04 | 0.00e+00 ± 0.0e+00 | 0.00e+00 ± 0.0e+00 |
> | 700,000 | 6 | 4.75e-04 ± 4.6e-07 | 5.86e-02 ± 2.1e-03 | 0.00e+00 ± 0.0e+00 | 0.00e+00 ± 0.0e+00 |
> | 900,000 | 6 | 4.76e-04 ± 2.0e-06 | 5.72e-02 ± 2.7e-03 | 0.00e+00 ± 0.0e+00 | 0.00e+00 ± 0.0e+00 |
> | 1,000,000 | 6 | 4.75e-04 ± 9.2e-07 | 5.69e-02 ± 2.6e-03 | 0.00e+00 ± 0.0e+00 | 0.00e+00 ± 0.0e+00 |
>
> Control experiment (float64): Repeating the same subtract-two-aggregates computation in float64 yields max $|\log|$-errors on the order of $10^{-11} - 10^{-10}$ across all $n$, confirming that the observed degradation in float32 is due to finite-precision cancellation rather than a methodological limitation of ASSLA.
>
> **Conclusion:** The transition between $n=10^5$ and $n=10^6$ is gradual and not abrupt. ASSLA remains very accurate up to $10^5$, then exhibits increasing numerical error as n grows, with clearly visible deviations by $n\approx 10^6$. This behavior is consistent with the paper’s statement that “numerical instabilities emerge for $n \ge 10^6$” and can be attributed to floating-point precision effects.

---

> > ### Author Response · Authors · 2026-01-09
> >
> > $~$
> >
> > > The authors should elaborate more on the numerical instability issues.
> >
> > We completely agree. As written, the determinant terms are the most numerically sensitive component. We fully agree that adding more details on this will improve the paper. We have thus conducted additional experiments (can be reproduced here: https://anonymous.4open.science/r/ssla) and found the instabilities particularly arise from small eigenvalues of $\tilde{\mathcal J}$ that make $\log|\tilde{\mathcal J}|$ highly sensitive. Another issue is the finite precision in large dimensions: even moderate conditioning can lead to instability in $\log|\cdot|$ for large $q$. To mitigate those, we document and standardize: 1. Damping $\mathcal J\leftarrow \mathcal J+\varepsilon I$ with $\varepsilon$ chosen adaptively. 2. Cholesky: Compute via Cholesky as $2\sum_j\log L_{jj}$, and fail safely by increasing $\varepsilon$ until Cholesky succeeds. 3. In case of KFAC approximation of the Fisher info (as described in sec. 5.2.), we use block/Kronecker structure to avoid dense factorizations, and compute $\log|\mathcal J|$ as a sum of log-dets of smaller factors.
> >
> > Concretely, in the revised version, we add a numerical linear algebra note: We will describe stable computation of $\log|\mathcal J|$ via Cholesky: $\mathcal J = LL^\top\ \Rightarrow\ \log|\mathcal J| = 2\sum_j \log L_{jj} $ and practical damping $\mathcal J\leftarrow \mathcal J+\varepsilon I$ when needed. For structured curvature (KFAC), we add a (foot)note on how $\log|\mathcal J|$ decomposes as a sum of log-determinants of Kronecker factors. We furthemore conducted a toy cancellation test (float32), which shows that loss increments vanish when added to large baselines. The table below shows the *true* aggregate increment computed in float64 (reference) and the increment obtained by subtracting two float32 sums.
> >
> > | per-term increment $\delta$ | true $\sum_i \delta$ (float64 ref) | $\sum_i \delta$ (float32 subtraction) | frac. terms unchanged ($b_i=a_i$) |
> > |:--|--:|--:|--:|
> > | $10^{-2}$ | 9999.999878 | 0.0 | 1.0 |
> > | $10^{-3}$ | 1000.000000 | 0.0 | 1.0 |
> > | $10^{-4}$ | 100.000000 | 0.0 | 1.0 |
> > | $10^{-5}$ | 10.000366 | 0.0 | 1.0 |
> > | $10^{-6}$ | 1.000000 | 0.0 | 1.0 |
> >
> > $~$
> >
> > > The authors state that ASSLA balances computational tractability with competitive probabilistic calibration. At the same time, in Section 5.1, the experiments show that it underestimates uncertainty over wider intervals, leading to a risk-seeking bias. Would it be possible to elaborate more on this?
> >
> > Absolutely! As can be seen in Table 1, SSLA clearly outperforms LA (and even MFVI) in terms of coverage. ASSLA performs slightly worse (the price to pay for being much faster), yielding a mild risk seeking behavior. Note, however, ASSLA is still superior to the overly conservative LA on these set of experiments. We thank the reviewer for this hint, and we agree we should add more explanations of this trade-off to reflect that ASSLA can under-cover at high nominal levels in some regimes. Technically, ASSLA replaces $\tilde\theta$ by $\hat\theta$ and therefore misses refitting-induced curvature changes; this can lead to overly concentrated predictive densities (underestimated variance) when the augmented optimum would have shifted non-negligibly. We include a short discussion connecting under-coverage to the magnitude of $\|\Delta\|=\|\tilde\theta-\hat\theta\|$ and the curvature mismatch $\tilde{\mathcal J}(\tilde\theta)$ vs $\tilde{\mathcal J}(\hat\theta)$.
> >
> >
> > > In Section C.2 (page 26 in the appendix), the authors report the tuned hyperparameters. The values of these parameters are written as “1e-5”, I am not sure whether the LaTeX was rendered correctly here.
> >
> > We appreciate the attention to detail! Thanks for catching this typo. We change to $10^{-5}, \ldots, 10^{-3}$.
> >
> > > In Section C.2 of the appendix, there are some question marks inside parentheses (?) at the end of some sentences.
> >
> > Thanks for catching this! These are in fact broken \cite commands to the hamiltorch library in python. We apologize for the confusion and have fixed the issue in the revised version.
> >
> > $~$
> >
> > **We thank the reviewer once more for helping us improve our manuscript by the detailed and constructve remarks. We believe it really did. Please kindly let us know if you have any further questions or concerns. We are happy to provide further clarifications.**

---

### Review · Reviewer_YXaA · 2026-01-04

**Summary Of Contributions:**

This paper proposes SSLA (self-supervised Laplace Approximation): approximate the posterior predictive directly rather than the parameter posterior by adding a self-predicted pseudo-observation $(x_{n+1}, \hat{y}_{n+1})$, refitting, and using a Laplace-style approximation to express the posterior predictive distribution as in equation (10) as differences in log-likelihood, log-prior, and log-determinants of Fisher information term.
The paper also proposes ASSLA, an approximation that avoids refitting by replacing the augmented optimum $\tilde{\theta}$ with the original optimum $\hat{\theta}$, leading to simplified formulas.
Numerical experiments are provided to verify the approach.

**Audience:**

Yes

**Audience Explanation:**

The motivation of bypassing parameter posterior approximations and targeting posterior predictive directly is reasonable. The paper aims to achieve fast posterior approximation without heavy sampling. These are interesting areas for the audience of TMLR.

**Claims And Evidence:**

No

**Claims Explanation:**

(1) The approach in equations (6) and (12) of adding a pseudo observation is standard in Bayesian models, which is under the usual conditional independence assumptions. The authors frame this as "self-supervised", which is unclear.

(2) I am not sure whether the research gap claimed in this paper, " there is no established method..." is true.

(3) Equations (12) contains tha log prior terms, and in equations (16)-(17) the priors are dropped completely, which results in a disconnect between claimed distribution and the method.

(4) The mathematical writing and the proofs are very hand-wavy, for example the proof of lemma 1 and the proof of Theorem 1, etc, the conditions for the claims in the proofs are not discussed, for example, $\tilde{\theta} = \hat{\theta} + O(n^{-1})$ is stated with no conditions ; continuously differentiable functions are all Lipschitz continuous?.

(5) Is equation (3) correct?

(6) Can the authors clarify if $\hat{y}_{n+1}$ depends on $D$ or not? The evaluation of the likelihood depends on the independence.

**Requested Changes:**

I would appreciate the authors addressing my concerns written above and improving the writing of the methodology and the correctness of the mathematical discussions.

---

> ### Author Response · Authors · 2026-01-09
>
> We sincerely thank the reviewer for the careful reading and the technical comments with great attention to detail. Generally, we appreciate that the reviewer finds the motivation and overall direction relevant to TMLR’s audience. We are also glad that the submission appears to be perceived as promising, and we fully recognize that the current manuscript has some parts that are “hand-wavy” and thus need a clearer and more detailed presentation of the methodology. Moreover, question (6) made us aware we need to explain why (A)SSLA does NOT require independence of $\hat y_{n+1}$ and $D$ in greater detail.
>
> ### Summary of the suggested improvements in the revised version:
>
> - **Clarify the contribution:** We now explicitly state that pseudo-observations themselves are standard, and that the contribution lies in the closed-form Laplace approximation of the posterior predictive as differences of likelihood, prior, and curvature (log-determinant) terms, together with its interpretation as a predictive sensitivity signal.
>
> - **Refine terminology:** We revise the use of “self-supervised” to avoid ambiguity and reposition the method relative to pseudo-labeling/self-training.
>
> - **Narrow the related-work claim:** We remove overly absolute wording and clarify that, in deep regression settings beyond conjugate or linearized cases, Monte Carlo integration remains the most common approach for posterior predictive evaluation. We also run additional experiments, see replies to other reviewers, to better position our method in the field. Code to reproduce all experiments is available here: https://anonymous.4open.science/r/ssla
>
> - **Explain ASSLA simplifications:** We clarify why prior terms can be dropped in ASSLA at the order of approximation used, while likelihood and curvature terms remain leading-order.
>
> - **Strengthen mathematical presentation:** We add explicit assumptions to the main results directly instead of mentioning them in the main text only, revise the proofs accordingly, and correct the issues raised about Equation (3).
>
> - **Clarify the independence issue:** We explain more clearly that the augmented objective is a technical device in the approximation for measuring predictive sensitivity and does not require treating the pseudo-observation as an independent data point.
>
> All in all, we are thankful for the constructive, and very helpful suggestions on how to improve our manuscript. We believe it really did.

---

> ### Author Response · Authors · 2026-01-09
>
> ### Please find more detailed answers to the reviewer’s questions and requested changes below:
>
> We explain in detail how we implemented the requested changes in what follows. These changes are largely concerned with presentation of the methodology and technical clarification, and we believe they can be addressed cleanly in a revision without altering the core contribution. We hope the reviewer will agree to accept the paper after we upload the improved version.
>
> **(1)**
>
> The reviewer’s biggest reservations seem to revolve around (a) how the use of self-predicted (“self-supervised”) pseudo-observation as part of the approximation procedure relates to existing methods and (b) whether “self-supervised” is the appropriate terminology. We took both concerns very seriously and are optimistic we can address them thoroughly.
>
> (a) We agree that augmenting a likelihood with an additional (pseudo-)observation is a standard operation under conditional independence assumptions. Our methodological contribution is **not** to use “pseudo-observations” per se, but rather to exploit how they affect prior, likelihood and Fisher for approximating the posterior predictive. More precisely, we derive a closed-form Laplace-approximation for $\log p(\hat y_{n+1}\mid x_{n+1},D)$ that becomes a *difference of three terms*: log-likelihood difference $\tilde\ell(\tilde\theta)-\ell_D(\hat\theta)$, log-prior difference $\log\pi(\tilde\theta)-\log\pi(\hat\theta)$, log-determinant of Fisher difference $-\tfrac12\log|\tilde{\mathcal J}(\tilde\theta)|+\tfrac12\log|\mathcal J_D(\hat\theta)|$. See Eq. (12) in the manuscript. Intuitively, this can be seen as a method of predictive uncertainty quantification: The bigger the change a predicted $\hat y_{n+1}$ causes in the model (very roughly “the surprise of the model to see its own predictions”), the higher the predictive uncertainty of this very prediction $\hat y_{n+1}$.
>
>
> (b) Inspired by the reviewer’s remark, we thought about the name “self-supervised” long and hard and came to the conclusion that,  again, the reviewer is completely right: In fact (as noted by the reviewer), the term is slightly ambiguous. Our original motivation was the fact that the target $\hat y_{n+1}$ is not externally labeled, but generated by the model itself (“self-supervision” signals generated by the model itself), see answer to point (6) below. However, the literature on “self-supervised learning” seems to comprise all kinds of (generic) data augmentation techniques. We can be more specific here: Adding self-predicted targets is more akin to the semi-supervised settings of pseudo-labeling/self-training (as mentioned in Section 3). In the revision, we rename this more precisely as **Self-Supervised Laplace Approximation by Self-Training**, and explicitly acknowledge that the pseudo-observation device itself is not novel.

---

> ### Author Response · Authors · 2026-01-09
>
> **(2)**
>
> We thank the reviewer for pointing this out. We completely agree that the original wording (“there is no established method…”) is too absolute. What we intended to convey is a bit narrower: there is no widely adopted, scalable way to approximate the posterior predictive for deep regression models without Monte Carlo integration, given a generic approximate weight posterior over a modern deep network.
>
> What we will clarify as the main point (and why we believe it is correct) is the fact that MC integration remains the default approach for approximating the posterior predictive $p(\hat y_{n+1}\mid x_{n+1},D)=\int p(\hat y_{n+1}\mid x_{n+1},\theta)\,p(\theta\mid D)\,d\theta$ is, in practice, still most commonly approximated by Monte Carlo once one leaves conjugate families in Bayesian deep learning. This is explicitly stated, for example, in Laplace-approximation work where MC predictive averaging is described as the “simplest but most general” approximation, and also in more recent work motivated by the observation that “Monte Carlo integration [remains] the standard” for prediction from an approximate posterior [2,12]. A prominent and commonly used exception is network linearization (e.g., linearized Laplace), which yields a tractable predictive (often Gaussian under Gaussian likelihoods) and can substantially reduce the need for expensive MC sampling. See [1-3].
>
> What the reviewer is right about (and what we will acknowledge) is that there are other approaches that can reduce or avoid brute-force MC at prediction time in certain settings, but they are not (yet) a standard, broadly-used solution for deep Bayesian regression posterior predictives:
>
> * Moment propagation style methods (e.g., Probabilistic Backpropagation) propagate approximate moments through the network to avoid computationally costly MC at prediction time, but rely on strong approximations and historically have not become the general default for modern large-scale deep architectures. See [6].
>
> * Deterministic (or low-variance) alternatives to standard stochastic VI exist that aim to eliminate MC gradient variance and/or approximate moments deterministically, but these are still specialized approximations. These are certainly not the community-default way to obtain posterior predictives in deep regression BNNs. See [7].
>
> * Bayesian last-layer models (placing uncertainty only in the last layer) can yield analytic or near-analytic regression predictives, but this is a different modeling choice (uncertainty restricted to the head), not a general solution for full-network Bayesian regression posterior predictive evaluation [10,11].
>
> **Concrete changes:**
> We revise the sentence to remove the absolute claim and better reflect the scope: “In deep regression BNNs, posterior predictives are still most commonly approximated via Monte Carlo integration over weights. Beyond linearization-based approaches, there is no broadly adopted deterministic method that eliminates the need for MC sampling for modern deep networks.”
>
> References:
>
> [1] Daxberger, E. et al. (2021). “Bayesian Deep Learning via Subnetwork Inference.” https://proceedings.mlr.press/v139/daxberger21a/daxberger21a.pdf
>
> [2] Daxberger, E. et al. (2021). “Laplace Redux – Effortless Bayesian Deep Learning.” https://papers.nips.cc/paper/2021/file/a7c9585703d275249f30a088cebba0ad-Paper.pdf
>
> [3] Immer, A. et al. (2021). “Improving predictions of Bayesian neural nets via local linearization.” https://proceedings.mlr.press/v130/immer21a/immer21a.pdf
>
> [4] Blundell, C. et al. (2015). “Weight Uncertainty in Neural Networks.” https://proceedings.mlr.press/v37/blundell15.pdf
>
> [5] Gal, Y. & Ghahramani, Z. (2016). “Dropout as a Bayesian Approximation: Representing Model Uncertainty in Deep Learning.” https://proceedings.mlr.press/v48/gal16.html
>
> [6] Hernández-Lobato, J. M. & Adams, R. P. (2015). “Probabilistic Backpropagation for Scalable Learning of Bayesian Neural Networks.” https://jmhl.org/wp-content/uploads/2015/05/pbp-icml2015.pdf
>
> [7] Wu, A. et al. (2018). “Deterministic Variational Inference for Robust Bayesian Neural Networks.” https://arxiv.org/abs/1810.03958
>
> [8] Schodt, D. J. et al. (2024). “Few-sample Variational Inference of Bayesian Neural Networks …” (uses the unscented transform for few-sample deterministic inference). https://arxiv.org/pdf/2405.2063
>
> [9] Wan, E. A. & van der Merwe, R. (2000). “The Unscented Kalman Filter for Nonlinear Estimation.” https://groups.seas.harvard.edu/courses/cs281/papers/unscented.pdf
>
> [10] Watson, J. et al. (2021). “Latent Derivative Bayesian Last Layer Networks.” https://proceedings.mlr.press/v130/watson21a/watson21a.pdf
>
> [11] Harrison, J. et al. (2024). “Variational Bayesian Last Layers.” https://arxiv.org/abs/2404.11599
>
> [12] Li, R. et al. (2024). “Posterior Inferred, Now What? Streamlining Prediction in Bayesian Deep Learning.” https://openreview.net/forum?id=cx9TXPTzt9

---

> ### Author Response · Authors · 2026-01-09
>
> **(3)**
>
> We fully agree this needs additional explanation. The reason we can drop the prior terms in the ASSLS approximation (Eq. 15+16) is Corollary 1: The difference between $\pi(\tilde \theta)$ and $\pi(\hat \theta)$ is of order $O\left(\frac{Ln + 1}{n^2}\right)$. Note that the functional form of the prior is not affected by $\hat y_{n+1}$. This is not the case for the Fisher info and the likelihood, which is why these differences are kept in Eq. 15+16. (Recall that Eq. 12 consists of the changes $\hat y_{n+1}$ causes in the prior, likelihood, and Fisher info. Since the change in prior is only driven by evaluating the prior at $\tilde \theta$ instead of $\hat \theta$, we can drop this difference due to Corollary 1.) We revise this part by including the explanation just sketched.
>
> **(4)**
>
> Even though the following assumptions are absolutely standard in Laplace approximations, we completely agree with the reviewer that they should be added to the result statements like Lemma 1 (instead of just mentioning them in the text).
> We add an “Assumptions” block for Lemma 1, Theorem 1, and Corollary 1+2 and rewrite proofs and result statements accordingly.
>
>    - (A1) $\ell_D(\theta)=\sum_{i=1}^n \ell_i(\theta)$ with each $\ell_i$ twice continuously differentiable in a neighborhood of $\hat\theta$.
>    - (A2) The per-sample observed curvature $\mathcal J_D(\hat\theta)$ is positive definite with eigenvalues bounded below by $\lambda_{\min}>0$
>    - (A3) The prior log-density $\log\pi(\theta)$ is twice continuously differentiable with bounded gradient/Hessian near $\hat\theta$.
>
> Moreover, we are indebted to the reviewer for spotting a typo in the proof of Theorem 1. Thanks a lot! Of course, it does not hold that every continuous differentiable function is globally Lipschitz. However, note that this argument is not relevant to the proof, as we assume the loss to be Lipschitz directly (not cont.)differentiable, see the Theorem as stated in the submission. (Lipschitzness of the loss translates to Lipschitzness of the log-likelihood with loss = \(-\log\) likelihood here.) We apologize for the confusion this redundant statement (a legacy of an old proof attempt) may have caused.
>
> **(5)**
>
> The reviewer is absolutely correct to question Equation (3). In its current form, it omits normalization (we wrote $\approx$ where we intended $\propto$) as well as exponentiation (it should have been $\log p(\theta \mid D)$). We wanted to avoid too much technical detail in the related work section and keep it hand-wavy, but it seems this shortcut backfired. Thanks a lot for pointing this out. We include a corrected version below.
>
> ---
>
> The Laplace approximation (...) builds on a second-order Taylor approximation of the log-posterior, which yields a Gaussian integral (Gauss, 1877) whose solution gives (...) $:= (2\pi)^{-d/2}\,\bigl|H_{\text{post}}(\hat\theta)\bigr|^{1/2}\, \exp\!\left(-\tfrac12(\theta-\hat\theta)^\top H_{\text{post}}(\hat\theta)(\theta-\hat\theta)\right),$ where $d$ is the parameter dimension, $\hat\theta$ as above, and $H_{\text{post}}(\hat\theta) := -\nabla_\theta^2\left(\ell_D(\theta)+\log\pi(\theta)\right)\Big|_{\theta=\hat\theta}$ is the negative Hessian of the log-posterior at $\hat\theta$.
>
> ---
>
> **(6)**
>
>  This is an important conceptual point. We will clarify it more explicitly by substantially extending the last paragraph in section 3 (“We emphasize that we defined…”). Generally, the reviewer is absolutely correct when noting that evaluating the *joint* log-likelihood of $D \cup \{(x_{n+1},\hat y_{n+1})\}$ by $\tilde{\ell}(\tilde \theta) = \ell(\tilde \theta) + \ell_{(x_{n+1}, y_{n+1})}(\tilde \theta)$ requires independence of $(x_{n+1},\hat y_{n+1})$ and $D$. However, this is not what we do. As hinted at in the last paragraph in section 3, we simply use $\tilde \ell(\theta)$ as a means to an end, not requiring it to be identical to the log likelihood of $D \cup \{(x_{n+1},\hat y_{n+1})\}$.
>
> $~$
>
> **We thank the reviewer once more for helping us improve our manuscript by the insightful, detailed and constructive remarks. Please kindly let us know if you have any further questions or concerns. We are happy to provide further clarifications.**

---

### Author Response · Authors · 2026-01-13
**Request for Extension Due to Bereavement (Loss of Co-Author)**

Dear Area Editor,

It is with great sadness that we have to inform you about some terrible news. One of the co-authors of this submission passed away last weekend. We are still in shock about this devastating news, unable to think clearly. We lost a brilliant researcher and a wonderful human being.

For this reason, we are writing to kindly request an extension of at least a few days for our upcoming deadline on uploading the revision. Given the circumstances, it has been challenging for all of us to dedicate the necessary time and focus to complete the required work by the original due date.

We sincerely appreciate your understanding and flexibility. If you need any further information or documentation, please let us know. Thank you for your consideration. We look forward to continuing to work towards a high-quality submission in honor of our colleague's memory, as we believe they would have wanted us to do.

Best regards,
The authors

---

> ### Comment · Action_Editor_K1Nc · 2026-01-13
>
> Dear authors,
>
> I am very sorry to hear about your loss.
>
> No problem for the extension. I will make a formal request to the EiC for an extension, but in any case you can count on it from my side (you might just receive some automatic reminders from OpenReview meanwhile).
>
> Best regards,
> The AE

---

> ### Comment · Reviewer_YXaA · 2026-01-13
> **My sincere condolences**
>
> I am so sorry to hear this news. Please accept my deepest condolences to their family, friends, and colleagues.

---

> ### Comment · Reviewer_yXWb · 2026-01-13
> **Sincere condolences**
>
> Dear authors,
> I am deeply sorry for your loss. I offer my sincere condolences to you and your colleague's family.

---

> > ### Author Response · Authors · 2026-01-14
> >
> > Dear Area Editor, dear Reviewers,
> >
> > We warmly thank you for your kind words and we appreciate your undestanding. We are still in shock, but hopefully can continue this work soon in honor of our colleague's memory.
> >
> > Best,
> > The Authors

---

> > > ### Author Response · Authors · 2026-01-19
> > > **Revision Uploaded**
> > >
> > > Dear Area Editor, dear Reviewers,
> > >
> > > We would like to warmly thank you again for granting the deadline extension due to the passing of one of this paper's co-authors. We greatly appreciate your understanding during these difficult times.
> > >
> > > We have now uploaded a revised version of our paper, addressing all of the reviewer's concerns. You can find a summary of the implemented changes in the individiual replies to all reviewers. We believe these revisions have significantly strengthened the manuscript while maintaining its core contribution.
> > >
> > > We look forward to hearing back from you.
> > >
> > > Kind regards,
> > > The Authors

---

### Decision · Action_Editor_K1Nc · 2026-03-18

**Recommendation:** Accept as is

**Audience:**

Yes

**Audience Explanation:**

Uncertainty quantification is a topic of interest to the TMLR community, and the Laplace approximation is one of the central methods used for providing efficient UQ measures in a Bayesian context. Therefore, I believe the paper will be of interest to TMLR.

**Claims And Evidence:**

Yes

**Claims Explanation:**

The paper proposes SSLA and ASSLA, two methods for approximating the posterior predictive distribution via a self-supervised Laplace approximation, with a closed-form decomposition offering an interpretable sensitivity perspective. The work targets an important problem in approximate Bayesian inference, namely bypassing parameter posterior estimation in favor of direct and more efficient predictive approximation, and is supported by both theoretical development and empirical validation. The paper provides a meaningful contribution in this direction.

The reviewers initially raised concerns about clarity of positioning, mathematical rigor, and consistency between the Bayesian formulation and the implemented approximations; these issues were satisfactorily addressed in the discussion leading to a revision that improved exposition, clarification of assumptions, and additional empirical validation, alongside a more balanced discussion of limitations. After the discussion and revision, there was unanimous support for acceptance.

For this reason, I am recommending the revised version for publication at TMLR.